# Minute-scale control of ubiquitin-mediated degradation reveals dynamics of bacterial secreted effector-functions

Haolin Zhang(张浩霖) [1], Yongxia Guo[1], Bikash Adhikari [2], Nevenka Dudvarski-Stankovic[2], Elmar Wolf [2] & Thomas Rudel [1] ✉

Precise temporal control of protein abundance is essential for dissecting dynamic cellular processes. While degron-based systems enable rapid protein depletion in eukaryotic cells, comparable tools are lacking for bacterial effectors delivered into host cells during infection. Here, we establish AIDE (Auxin-Inducible Degradation of Effectors), a host-directed degradation platform that harnesses the ubiquitin-proteasome system to selectively eliminate secreted bacterial proteins, including membrane-integrated effectors. By integrating a minimal auxin-inducible degron (AID) tag into effector genes, AIDE enables rapid, reversible, and spatially confined degradation while preserving native expression and secretion. We apply AIDE to *Chlamydia trachomatis* and show, that the membrane-integrated deubiquitinase Cdu1 suppresses autophagy early and later promotes developmental transitions, whereas the integral membrane fusogen IncA remains continuously required to maintain homotypic inclusion fusion. This AIDE platform provides minute-scale, spatiotemporal control over bacterial effector activity and offers a broadly applicable framework for dissecting virulence mechanisms and host-pathogen interactions across diverse secretion-dependent pathogens.

Bacterial infections are the second leading cause of death worldwide[1]. Many pathogenic bacteria cause diseases by delivering proteins into host cells via specialized secretion systems. Among these, the bacterial type III secretion system (T3SS) acts as a protein transport apparatus that numerous Gram-negative bacteria use to inject effector proteins directly into host cells. Also known as an injectisome, the T3SS functions like a nanosyringe enabling bacteria to deliver proteins straight into the host cytoplasm or organelles, bypassing the extracellular space[2,3]. These effector proteins manipulate host cell processes to facilitate infection[4,5]. The ability to cause disease in many clinically important bacterial species relies on a functional injectisome. This group spans a wide spectrum of pathogens, including food- and waterborne bacteria, such as enteropathogenic *Escherichia coli*, *Salmonella enterica* serovar Typhimurium, and *Shigella dysenteriae*; those spread by insect vectors or animal hosts, including *Rickettsia* species

and *Yersinia pestis*; healthcare-associated pathogens like *Pseudomonas aeruginosa*; and bacteria responsible for sexually transmitted infections, such as *C. trachomatis*[2,6,7]. T3SSs are highly dynamic structures that inject their substrates into host cells within minutes[8]. Many of the secreted effector proteins have enzymatic or structural functions, and researchers traditionally study them either by loss-of-function approaches in the pathogenic bacteria or by overexpression in the host cell of interest[9]. Both strategies have limitations: loss-of-function screens are slow and carry the risk of compensatory events arising during mutant selection, while overexpression in host cells may produce artifacts. Similarly, overexpression of bacterial effectors in host cells often leads to their mislocalization to inappropriate cellular compartments. Moreover, unphysiological levels of effectors can create artifacts by disrupting the balance of host cell processes. For many obligate intracellular bacteria, effectors may be essential for

[1]Chair of Microbiology, University of Wuerzburg, Wuerzburg, Germany. [2]Institute of Biochemistry, University of Kiel, Kiel, Germany.
✉e-mail: thomas.rudel@uni-wuerzburg.de

bacterial survival or replication, so gene knockouts can compromise pathogen viability[10,11].

A strategy that overcomes these limitations is the rapid and controllable degradation of secreted bacterial effectors within host cells. In eukaryotes, targeted protein degradation (TPD) emerges as a therapeutic approach to eliminate specific proteins[12,13]. A prominent example is the use of bifunctional small molecules, known as PROTACs, which recruit target proteins to E3 ubiquitin ligases for proteasomal degradation. However, PROTACs development depends on the presence of druggable domains within the target protein, requires extensive medicinal chemistry optimization, and often fails[14,15]. An alternative approach uses genetic degron systems, which fuse ligand-responsive tags to proteins of interest and enable controlled degradation[16–19].

The auxin-inducible degron (AID) system exploits the plant F-box receptor TIR1, a substrate adapter of the SCF E3 ligase complex (Skp1-Cullin1-Rbx1-TIR1). Upon auxin addition, the hormone binds to the leucine-rich repeat pocket of TIR1, promoting formation of a TIR1-auxin-AID ternary complex with the degron-tagged substrate. This interaction triggers K48-linked polyubiquitination and subsequent 26S proteasome-mediated degradation of the target protein. In most cases, auxin treatment depletes tagged proteins within approximately 20-40 minutes[17]. The system was recently upgraded to the second-generation AID system (AID2), which uses engineered OsTIR1(F74G) and the synthetic auxin analog 5-Ph-IAA in a "bump-and-hole" strategy. This modification virtually eliminates background degradation and dramatically increases ligand sensitivity, while further accelerating degradation kinetics[20]. With high specificity and efficiency, AID2 allows precise, reversible protein depletion, making it ideal for studying dynamic cellular processes.

Here we develop a platform for auxin-inducible degradation of effectors (AIDE). As a proof of principle, we focus on *Chlamydia trachomatis*, an obligate intracellular pathogen and the most common bacterial cause of sexually transmitted infections in humans[21]. *C. trachomatis* replicates within a membrane-bound compartment in the host cell, known as an inclusion, and follows a complex developmental cycle. Infectious elementary bodies (EBs) enter host cells and differentiate into replicative reticulate bodies (RBs). After several rounds of replication, RBs convert back into EBs, which exit the host cell to initiate new rounds of infection[22,23]. T3SS effectors are essential at all stages of chlamydial development, contributing to host cell entry[22,24], inclusion formation and stabilization[25–27], and the manipulation of multiple host signaling pathways to acquire nutrients and evade cell-autonomous defense mechanisms[22,28]. The chlamydial T3SS effector repertoire is estimated to comprise over 100 proteins, yet only a few have been functionally characterized[29,30]. Among these effectors, inclusion membrane proteins (Incs) are critical for pathogen survival. These proteins are embedded into the inclusion membrane and support the inclusion integrity and the development of *Chlamydia* by directly manipulating numerous host cell pathways.

The inducible direct degradation of effectors that we establish with the AIDE system enables investigations into the dynamics and function of secreted proteins that are not approachable by silencing technologies. These technologies work by inducible expression of regulatory RNAs[31] or the inducible expression of guide RNAs in CRISPR-expressing *Chlamydia*[32,33] to interfere with protein translation. Since these systems work by preventing new synthesis of proteins, they are used to determine the effect of inhibiting translation in general since a defined depletion of a protein in a short period of time to access the dynamics of function is not possible[34]. One reason for these limitations is the wide variation in protein half-lives, particularly the half-life of membrane proteins ranges from several hours up to days[35].

One of the Incs from *Chlamydia* particularly interesting to access by AIDE is the Chlamydia Deubiquitinase 1 (Cdu1) which interferes with inclusion ubiquitination, autophagy signaling and thereby supports

recruitment of Golgi-derived vesicles to the inclusion to support nutrient acquisition[36–38]. Cdu1 has two distinct enzymatic activities. Its deubiquitinase function removes ubiquitin tags, while its acetyltransferase activity acetylates itself and chlamydial effectors (e.g., IpaM, InaC) at lysine residues to protect them from ubiquitination[38]. Although defined mutants demonstrate both enzymatic activities of Cdu1, the temporal dynamics of these functions during infection, such as whether the inclusion remains ubiquitin-free for a defined period in the absence of Cdu1, remain unclear.

Another Inc protein whose temporal dynamics remain to be elucidated is IncA. By mimicking host SNARE proteins, IncA mediates inclusion fusion and prevents lysosomal degradation by competitively inhibiting interactions with host proteins such as Vamp3 and Vamp8[39,40]. Recent work indicates that fusion initiates at specialized inclusion contact sites (ICSs) - IncA- and lipid-enriched microdomains regulated by $PI(3,4)P_2$ and sphingolipids[40]. A key unresolved question central to the understanding of IncA's role is whether IncA acts exclusively to initiate inclusion fusion or is also required to maintain the fused-inclusion state over time.

Here, we implement the AIDE system to harness the host degradation machinery for the selective clearance of bacterial effectors at the host-*Chlamydia* interface, generating Ctr-AIDE. Using this platform, we dissect the temporal functions of Cdu1 and IncA, revealing their phase-specific roles in evasion of autophagy signaling, developmental transitions, and homotypic inclusion fusion. This study therefore provides a blueprint for the use of AIDE as a tool to analyze the dynamic functions of secreted effectors during bacterial infection.

## Results

### A genome-integrated platform for conditional protein depletion

To establish the AIDE system in *Chlamydia trachomatis* (*Ctr*), we designed a strategy to integrate degron sequences into chlamydial target effector proteins (Ctr-AIDE). Central to this system is a 7-kDa mAID degron, fused to target effectors and recognized by the mutant host F-box receptor OsTIR1(F74G), which forms the SCF E3 ligase complex and thus directs K48-linked ubiquitination upon addition of the synthetic auxin 5-Ph-IAA. To enable precise degron integration into *Ctr*, we combined AID2 with the FRAEM (Fluorescence-Reported Allelic Exchange Mutagenesis) system, a homologous recombination-based method for seamless genome editing[41]. This integration yielded Ctr-AIDE, a platform for rapid and reversible depletion of secreted effectors (Fig. 1A).

For several reasons, we selected the chlamydial deubiquitinase Cdu1 to establish the Ctr-AIDE strategy: (i) Cdu1 is a highly active K48 deubiquitinase that upon AIDE (that ubiquitinated K48 to induce degradation) could lower the efficiency of degradation; (ii) The auto-acetylation of Cdu1 at lysine residues has been implicated in ubiquitination resistance of the protein[38]; (iii) As a membrane integrated protein, degradation of Cdu1 is expected to be particularly challenging; (iv) Rapid Cdu1 protein degradation and re-expression could be a way to shed light on the effect of the deubiquitinating and acetyltransferase activities of Cdu1 in ongoing infections. To generate the Cdu1 degradation construct, a knock-in cassette was designed in the suicide plasmid pKW-L2[41], consisting of (i) 3 kb homology sequences flanking the *cdu1* locus; (ii) an mAID-FLAG sequence fused to the C-terminus of Cdu1 via a flexible linker (2x GGGS), ensuring cytosolic accessibility given that the Cdu1 C-terminus faces the host cytosol[36]; and (iii) a GFP selection cassette with spectinomycin resistance gene (aadA-GFP, Fig. 1B). This design enables allelic replacement of endogenous *cdu1* with the mAID-FLAG-tagged variant, ensuring native expression regulation and eliminating reliance on plasmid-based expression. *Ctr* recombinants were initially screened for loss of the plasmid pKW-L2 (RFP⁻) and the expression of GFP indicative of successful recombination. In addition, mAID tagging was confirmed by immunoblotting (Supplementary Fig. 1A). A matched Cdu1-FLAG

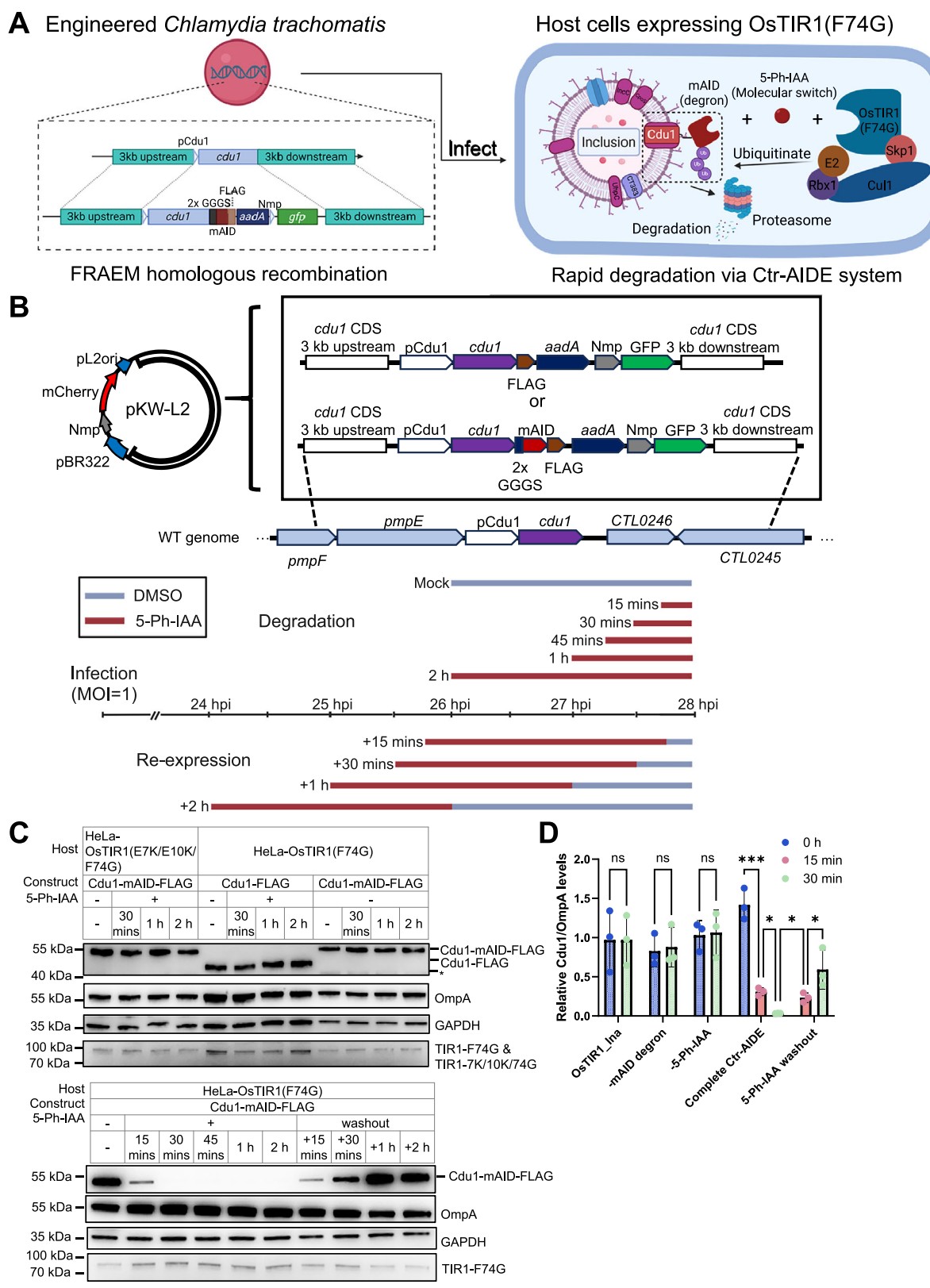

**A** Engineered *Chlamydia trachomatis*

Host cells expressing OsTIR1(F74G)

FRAEM homologous recombination

Rapid degradation via Ctr-AIDE system

**B**

control strain that lacks the mAID degron was constructed in parallel for comparative analysis (Supplementary Fig. 1A).

We established stable HeLa cell lines expressing either functional OsTIR1(F74G) or catalytically impaired OsTIR1(E7K/E10K/F74G)[42], and validated AID2 functionality by mAID-GFP degradation. The impaired OsTIR1(E7K/E10K/F74G) mutant served as a reliable degradation-negative control (Supplementary Fig. 2). Using this system, we

investigated Ctr-AIDE-mediated depletion of the effector Cdu1 (Fig. 1B). No K48 ubiquitin was detected on the inclusions in the Ctr-AIDE strain indicating that Cdu1-mAID functions in protecting the inclusion from ubiquitination. Strong K48 ubiquitination was visible on the inclusion already 5 min after addition of the 5-Ph-IAA inducer (Supplementary Fig. 3) suggesting that OsTIR1(F74G) is recruited to Cdu1-mAID and induces its ubiquitination. Surprisingly, strong

**Fig. 1 | AIDE system enables temporal control of Cdu1 expression in *Chlamydia*-infected HeLa cells. A** Schematic of Ctr-AIDE strategy, combining the FRAEM genome-editing platform (left) with the AID2 TPD system (right). Cdu1 is shown as proof of concept. Created in BioRender. Zhang, H. (2026) https://BioRender.com/t5v0ec7. **B** *Top*: Plasmid constructs for FRAEM-mediated homologous recombination in *Ctr* serovar L2. The targeted genomic region is marked by lines. pL2ori represents the putative chlamydial plasmid ori, and pBR322 (Ori) is included to facilitate cloning in *E. coli*. pCdu1, native Cdu1 promoter; Nmp, *Neisseria meningitidis* promoter; *aadA*, spectinomycin resistance gene; *Bottom*: Ctr-AIDE degradation/re-expression strategy. Bars indicate treatment with DMSO (control) or 5-Ph-IAA (1 μM). All samples were harvested at 28 hours post-infection (hpi) for downstream analysis. **C** Ctr-AIDE-mediated regulation of Cdu1-mAID. Cells were infected with indicated strains at MOI = 1. Cdu1 (anti-FLAG) and OsTIR1 mutants (TIR1-F74G

denotes OsTIR1(F74G)-9×Myc, and TIR1-7K/10 K/74 G denotes OsTIR1(E7K/E10K/F74G)-9×Myc; anti-Myc) levels were monitored, with Chlamydial Major Outer Membrane Protein (OmpA) as a loading control and GAPDH as a host cell control. Asterisks denote putative truncated or fragmented forms of Cdu1. **D** Quantification of Cdu1 degradation. FLAG signal (normalized to OmpA, mean ± SD) from 3 independent immunoblots (one shown in **C**). Statistical significance determined by two-tailed paired t-test (***$p < 0.001$, **$p < 0.01$, *$p < 0.05$; n.s., not significant, exact $p$ values were provided in Supplementary Data 3). OsTIR1_Ina, -mAID degron, -5-Ph-IAA: controls with dysfunctional E3 ligase OsTIR1(E7K/E10K/F74G), lacking mAID tagging, and lacking 5-Ph-IAA treatment, respectively. All experiments were performed ≥3 times with consistent results. Source data are provided as a Source Data file.

degradation of Cdu1-mAID was visible already 15 minutes after addition of the 5-Ph-IAA inducer and degradation was complete within 30 min (Fig. 1C, D). Degradation depended on the mAID tag since Cdu1 remained stable in (i) cells expressing the catalytically impaired OsTIR1(E7K/E10K/F74G) variant[42], (ii) strains expressing non-degron Cdu1-FLAG constructs, and (iii) untreated conditions. Importantly, degradation was fully reversible, with Cdu1-mAID re-accumulating partially within 15 min and completely within 2 h following 5-Ph-IAA washout after 2 h of treatment (Fig. 1C, D). Immunofluorescence microscopy confirmed rapid loss of inclusion membrane-localized Cdu1 after 5-Ph-IAA treatment and its restoration upon 5-Ph-IAA washout (Fig. 2A). Spatial resolution was further validated by 4 × expansion microscopy[43], which confirmed specific depletion of inclusion membrane-associated Cdu1 while pools in the chlamydial cytoplasm remained unaffected, highlighting the system's spatial precision (Fig. 2B). To assess target specificity of Ctr-AIDE for mAID-tagged effector, we monitored the untagged inclusion membrane protein IncA during rapid Cdu1 depletion. Immunoblotting showed that IncA expression remained unchanged while Cdu1-mAID was rapidly degraded (Supplementary Fig. 4), indicating that Ctr-AIDE selectively degrades mAID-tagged proteins.

Mechanistic dissection confirmed that Cdu1-mAID degradation depends on the host ubiquitin-proteasome pathways: degradation was blocked by the inhibition of the central Ubiquitin-activating enzyme 1 (TAK-243), and the proteasomal activity (MG-132, bortezomib). Interestingly, inhibition of the p97 ATPase with CB-5083 blocks Cdu1-mAID degradation. p97 functions by extracting misfolded or damaged proteins from membranes, a process triggered by ubiquitination[44], suggesting that degradation of inclusion membrane proteins involves p97, potentially through extraction of membrane proteins from the inclusion prior to proteasomal degradation. (Fig. 2C–E).

To assess the functionality of Ctr-AIDE in different cell lines, Cdu1 degradation was tested in A-375 (melanoma), HCT 116 (colon carcinoma), and U-2 OS (osteosarcoma). All lines exhibited degradation kinetics comparable to HeLa (cervical carcinoma) cells (Supplementary Fig. 5, 6), with HeLa showing the highest proteasomal degradation efficiency (Supplementary Fig. 5B). These results confirm that Ctr-AIDE functions robustly across diverse host cell types, with degradation fidelity dependent on the host ubiquitin-proteasome machinery rather than cell type-specific factors.

These results validate Ctr-AIDE as a genome-integrated AID2 platform that enables precise, conditional depletion of *Chlamydia* effectors across diverse host cell lines while preserving native expression dynamics. By eliminating plasmid dependencies and leveraging the host ubiquitin-proteasome pathways, Ctr-AIDE establishes a framework for dissecting essential bacterial virulence factors during infection.

## Ctr-AIDE functions in primary cells
To validate the broad applicability of Ctr-AIDE, we established the Ctr-AIDE system in primary cells. Primary Murine Reproductive Tract (PMRT) organoids, derived from whole female reproductive tract

digests, were transduced with lentivirus encoding OsTIR1(F74G) or OsTIR1(E7K/E10K/F74G) and selected for stable expression (Supplementary Fig. 7). Cells derived from these organoids were then grown as 2D monolayers to facilitate 5-Ph-IAA treatment when used for Ctr-AIDE degradation assays. In PMRT cells expressing OsTIR1(F74G), we observed rapid, 5-Ph-IAA-dependent depletion of mAID-tagged Cdu1 comparable to cancer cell lines: immunoblotting confirmed near-complete Cdu1 loss within 1 h of 5-Ph-IAA treatment, while controls (Primary cells expressing dysfunctional OsTIR1(E7K/E10K/F74G), DMSO treated cells, and non-degron Cdu1 constructs) showed no degradation (Fig. 3A, B). Immunofluorescence microscopy further demonstrated auxin-driven disappearance of inclusion membrane-localized Cdu1-mAID and its restoration within 2 h after 5-Ph-IAA washout (Fig. 3C). These results confirm that Ctr-AIDE operates robustly across both cancer and primary cell models, further prove its versatility for studying *Chlamydia* effector dynamics in physiologically relevant environments.

## Assessing Cdu1 Function Dynamics
Leveraging the Ctr-AIDE system, we first investigated the deubiquitinase function of Cdu1, previously implicated in countering autophagy signaling[36,37]. We focused on the autophagy marker p62 to monitor autophagy-associated labeling at inclusions and bypass potential confounding effects of Ctr-AIDE-induced K48-linked ubiquitination (Supplementary Fig. 3). Depletion of Cdu1 triggered delayed recruitment of autophagy marker p62 to inclusions: the p62 signal on inclusions was minimal 1 h after 5-Ph-IAA treatment but markedly increased by 4 h (Fig. 4A). This phenotype was rapidly reversible with p62 recruitment dissipating within 1 h of 5-Ph-IAA washout (Fig. 4A). Quantification of p62-positive inclusions confirmed this phenotype: Cdu1 depletion triggered p62 accumulation, while re-expression led to loss of p62 staining (Fig. 4B).

To further evaluate Cdu1's contribution to metabolic activity, and by extension overall fitness, we implemented two degradation strategies in the 2D PMRT cell model: (i) continuous degradation, in which 5-Ph-IAA treatment was initiated at 0, 8, 16, 24, 32, or 40 hpi and maintained until 48 hpi, and (ii) acute 8-hour pulses, initiated at staggered 8-hour intervals, and lasting 8 h each (Supplementary Fig. 8). Metabolic activity was monitored via RT-qPCR at 48 hpi for *gapA* (encoding glyceraldehyde-3-phosphate dehydrogenase), used here as a proxy for chlamydial metabolic activity under our assay conditions[45]. Sustained depletion significantly reduced *gapA* expression when initiated before 24 hpi (Fig. 4C). In contrast, short-term 8-hour degradation had no significant effect, indicating that Cdu1's metabolic role is only compromised by prolonged depletion (Fig. 4D). Consistent with previous reports[36,37], Cdu1 degradation did not impair metabolic activity in HeLa cells (Supplementary Fig. 9), by extension supporting the conclusion that Cdu1 is dispensable for chlamydial overall fitness in this cell type.

Cdu1 was also required for *Ctr* developmental progression[37]. During mid-late chlamydial developmental, bacteria transition from the non-infectious reticulate body (RB) to the infectious, non-

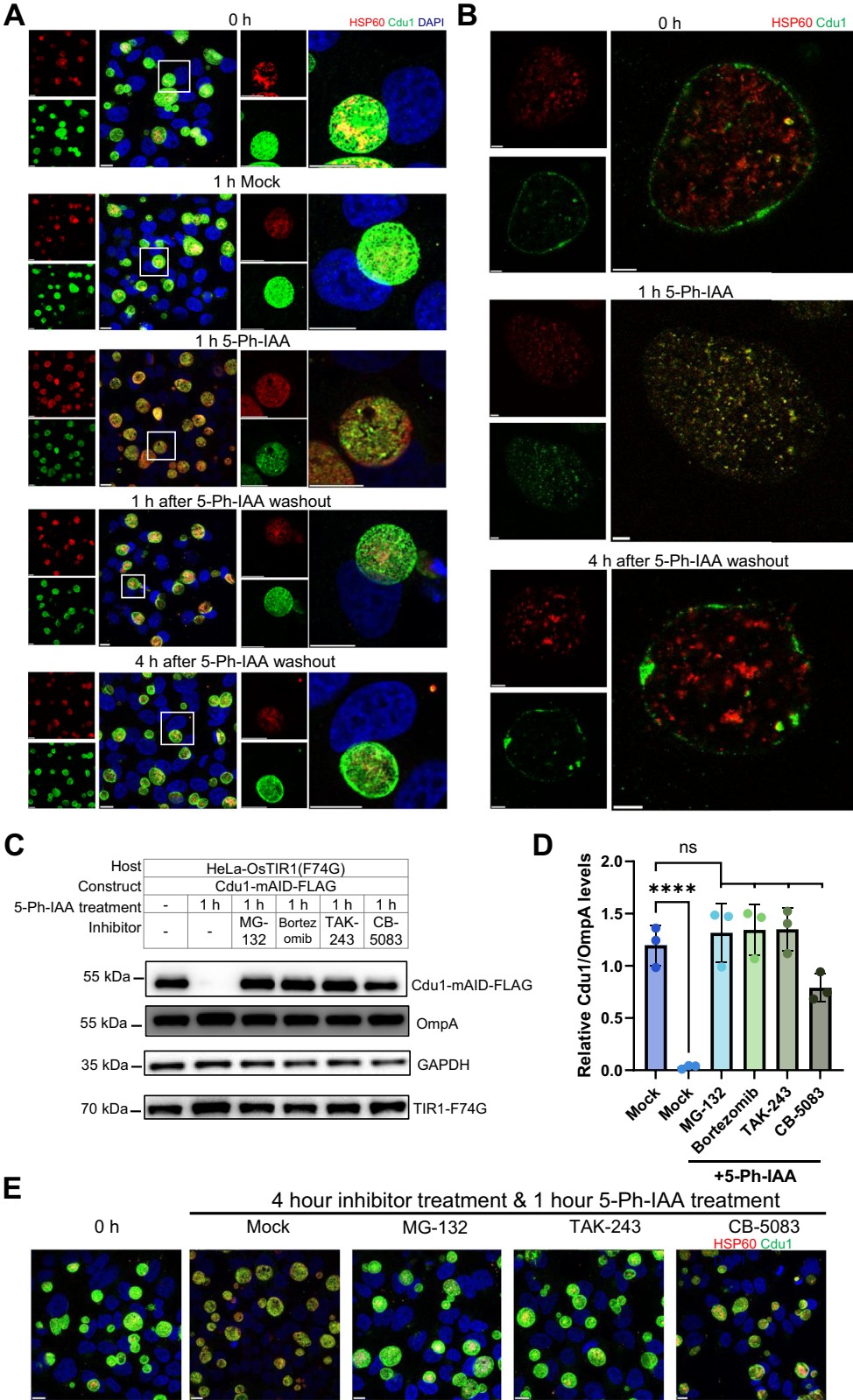

replicative elementary body (EB). Depletion before 24 hpi reduced levels of OmcB, a late-stage EB marker (Fig. 4E, F), coinciding with the onset of RB-to-EB differentiation[46]. These findings highlight Cdu1's critical role in initiating developmental transitions.

To assess the consequences of impaired EB formation, we evaluated progeny-associated infection burden using reinfection assays (Supplementary Fig. 10). Lysates from primary infected cells were used to infect fresh HeLa cells, and the extent of secondary infection was assessed by measuring OmpA levels at 24 hpi. OmpA (Major Outer Membrane Protein, MOMP) is highly expressed in EBs[47] and is critical for providing structural integrity to the chlamydial outer membrane. While OmpA abundance does not directly measure infectivity, it reflects bacterial load during the early stages of a new infection and thus serves as a surrogate readout of progeny-associated infection

**Fig. 2 | AIDE system enables spatial regulation of Cdu1 expression in *Chlamydia*-infected HeLa cells and depends on an intact ubiquitin-proteasome pathway. A** Immunofluorescence microscopy confirms Cdu1 depletion dynamics. Cdu1 (green, anti-FLAG), chlamydial inclusion (Hsp60, red), and DNA (DAPI, blue). The white square in the left panel marks the region enlarged in the right panel. Scale bar = 10 μm. **B** Expansion microscopy (4× expansion) resolving Cdu1 localization patterns under undegraded and degraded conditions. Scale bar = 10 μm. **C** Cdu1 degradation depends on ubiquitin-proteasome and p97 pathways. Host cells were pretreated with inhibitors for 3 h (including 1 hour during 5-Ph-IAA treatment) with pan-E1 (TAK-243, 1 μM), proteasome (MG-132, 10 μM; bortezomib, 1 μM), or p97 (CB-

5083, 10 μM) inhibitors. Degradation was blocked despite 1-hour 5-Ph-IAA exposure. **D** Quantification of inhibitor effects on Cdu1 degradation. FLAG levels (normalized to OmpA, mean ± SD) from three independent immunoblot replicates (one representative in **C**). Significance assessed by one-way ANOVA with Tukey's Multiple comparisons test. (****$p < 0.0001$; n.s., not significant, exact $p$ values were provided in Supplementary Data 3). **E** Immunofluorescence validation of inhibitors. Inclusion-localized Cdu1 signal persists in treated cultures. Scale bar = 10 μm. All experiments were replicated ≥3 times with consistent results. Source data are provided as a Source Data file.

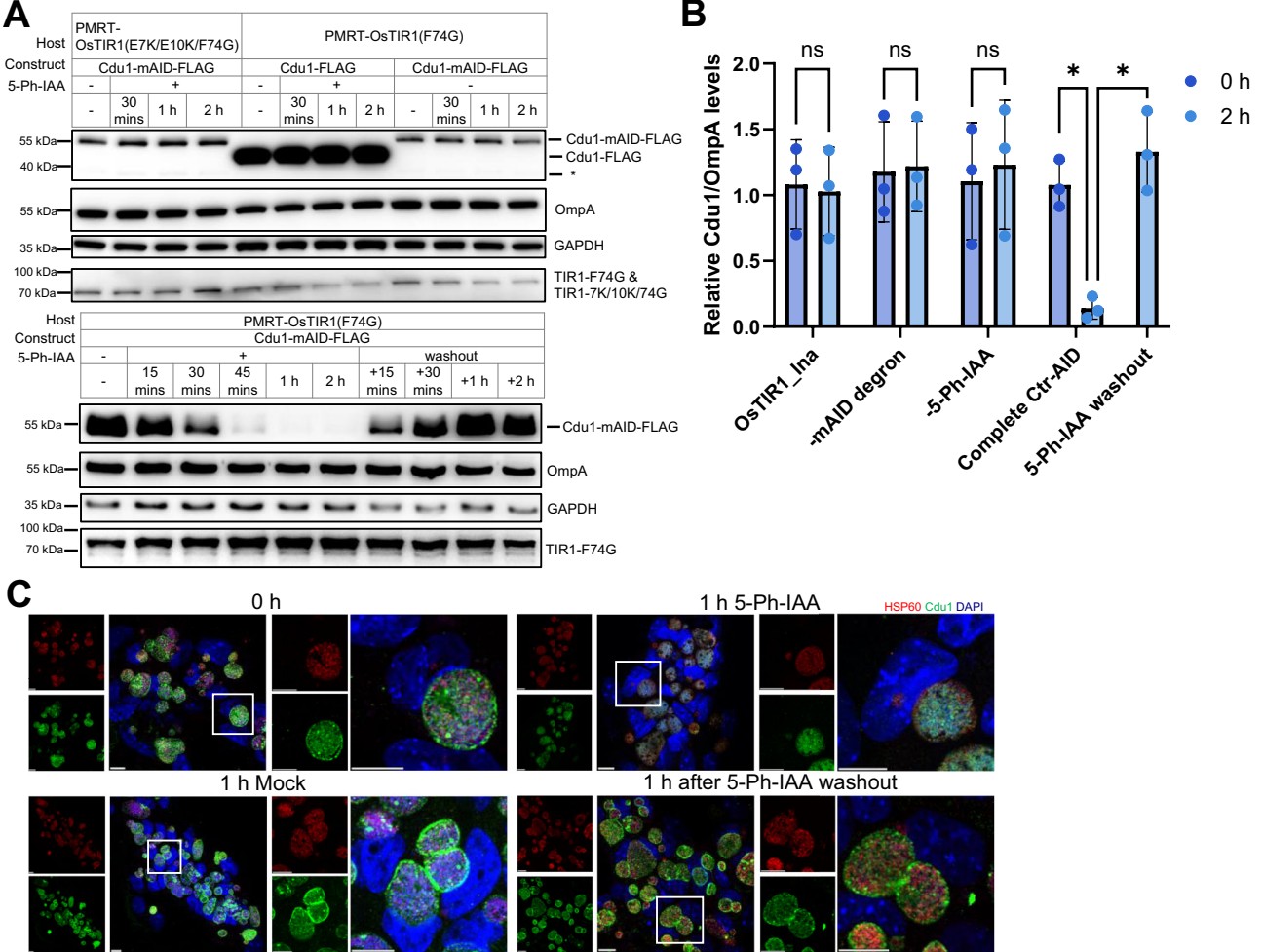

**Fig. 3 | Ctr-AIDE can be applied in primary cells. A** Immunoblot of the Ctr-AIDE-mediated regulation of Cdu1 levels in PMRT cells. Degradation treatment as in Fig. 2, but a higher infection load (MOI = 10). Cdu1 and OsTIR1 mutants (TIR1-F74G denotes OsTIR1(F74G)-9×Myc, and TIR1-7K/10 K/74 G denotes OsTIR1(E7K/E10K/F74G)-9×Myc) were detected with anti-FLAG and anti-Myc antibodies; OmpA and GAPDH served as loading controls. Asterisks denote putative truncated or fragmented forms of Cdu1. **B** Quantification of Cdu1 degradation in primary cells. OsTIR1_Ina, -mAID degron, -5-Ph-IAA: controls with dysfunctional E3 ligase OsTIR1(E7K/E10K/F74G), lacking mAID tagging, and lacking 5-Ph-IAA treatment,

respectively. FLAG levels (normalized to OmpA, mean ± SD) from three independent replicates (one shown in **A**). Significance assessed by two-tailed paired t-test (*$p < 0.05$; n.s., not significant, exact $p$ values were provided in Supplementary Data 3). **C** Fluorescence microscopy of Ctr-AIDE dynamics in primary cells. Cdu1 (green, anti-FLAG), inclusions (Hsp60, red), and DNA (DAPI, blue). The white square in the left panel marks the region enlarged in the right panel. Scale bar = 10 μm. All experiments were replicated ≥3 times with consistent results. Source data are provided as a Source Data file.

capacity. Cultures in which Cdu1 was depleted at or before 32 hpi yielded significantly reduced OmpA levels in secondary infections compared to untreated controls (Fig. 4G–I), consistent with impaired RB-to-EB differentiation in the primary culture. In contrast, no reduction in secondary infection burden was observed following acute 8-hour Cdu1 degradation pulses (Supplementary Fig. 11), reinforcing the requirement for sustained Cdu1 activity to support productive chlamydial development.

## Inducible degradation and re-expression of IncA

Previous studies demonstrated that the C-terminus of IncA mediates inclusion fusion[39]. To investigate the dynamics of IncA function as a key regulator of homotypic inclusion fusion[39,40], we applied the AIDE platform to this Inc protein. Using the same genome-integrated tagging strategy previously developed for Cdu1 (Supplementary Fig. 12), we generated *Chlamydia* strains expressing FLAG-tagged IncA or IncA-mAID-FLAG (Supplementary Fig. 1B).

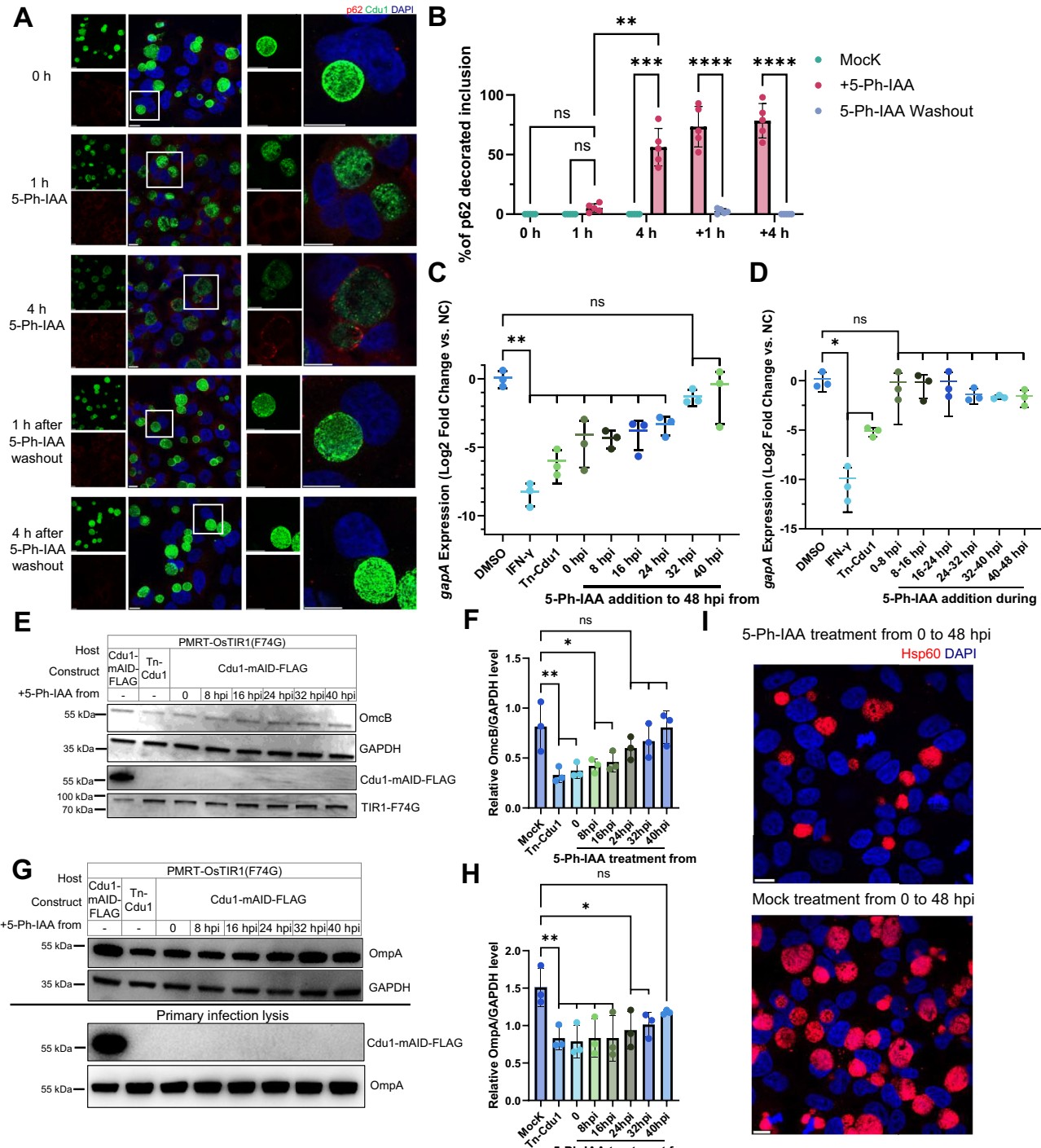

**Fig. 4 | Cdu1 loss induces transient p62 accumulation and impairs *Ctr* metabolic activity and progeny-associated secondary infection burden. A** p62 recruitment to inclusions upon Cdu1 depletion. p62 (red, anti-SQSTM1), Cdu1 (green, anti-FLAG), DNA (DAPI, blue). The white square in the left panel marks the region enlarged in the right panel. Scale bar = 10 μm. **B** Quantification of p62-positive inclusions. Data represent mean percentage ± SD from 500 inclusions per condition across five independent experiments. Statistical significance was assessed by two-way ANOVA with Tukey's test (****$p < 0.0001$, ***$p < 0.001$, **$p < 0.01$; n.s., not significant). +1 h and +4 h indicate hours after 5-Ph-IAA washout following 4 h treatment. **C** RT-qPCR of *gapA* (metabolic marker) and *rrs* (16S rRNA, control) at 48 hpi following timed 5-Ph-IAA treatment. DMSO treated samples, IFN-γ treated samples and Tn-Cdu1 strain served as controls[37]. Data are normalized to DMSO-treated controls and shown as mean ± SD from three independent biological replicates. Statistical analysis by one-way ANOVA with Dunnett's test (**$p < 0.01$; n.s.). **D** Short-term depletion shows no effect. Quantification and statistical analysis

were performed as in panel C.(*$p < 0.05$; n.s.). **E** Timed Cdu1 depletion disrupts RB-to-EB redifferentiation. OmcB (EB marker). Cdu1 (anti-FLAG), OsTIR1(F74G) (anti-Myc), GAPDH (loading control). **F** Quantification of OmcB normalized to GAPDH (mean ± SD) from three replicates (one shown in **E**). Statistical significance assessed by one-way ANOVA with Dunnett's test (**$p < 0.01$, *$p < 0.05$; n.s.). **G** Cdu1 loss reduces progeny-associated secondary infection burden. Lysates from primary infection (MOI = 1) were normalized by OmpA levels, then used to infect HeLa cells. **H** Quantification of OmpA abundance in secondary infections. OmpA was normalized to GAPDH (mean ± SD, n = 3; one replicate shown in **G**). Statistical analysis by one-way ANOVA with Dunnett's test (**$p < 0.01$, *$p < 0.05$; n.s.). **I** Immunofluorescence microscopy of secondary infections. Inclusions (Hsp60, red) and DNA (DAPI, blue). Scale bar = 10 μm. All experiments were replicated ≥3 times with consistent results. Exact *p* values were provided in Supplementary Data 3. Source data are provided as a Source Data file.

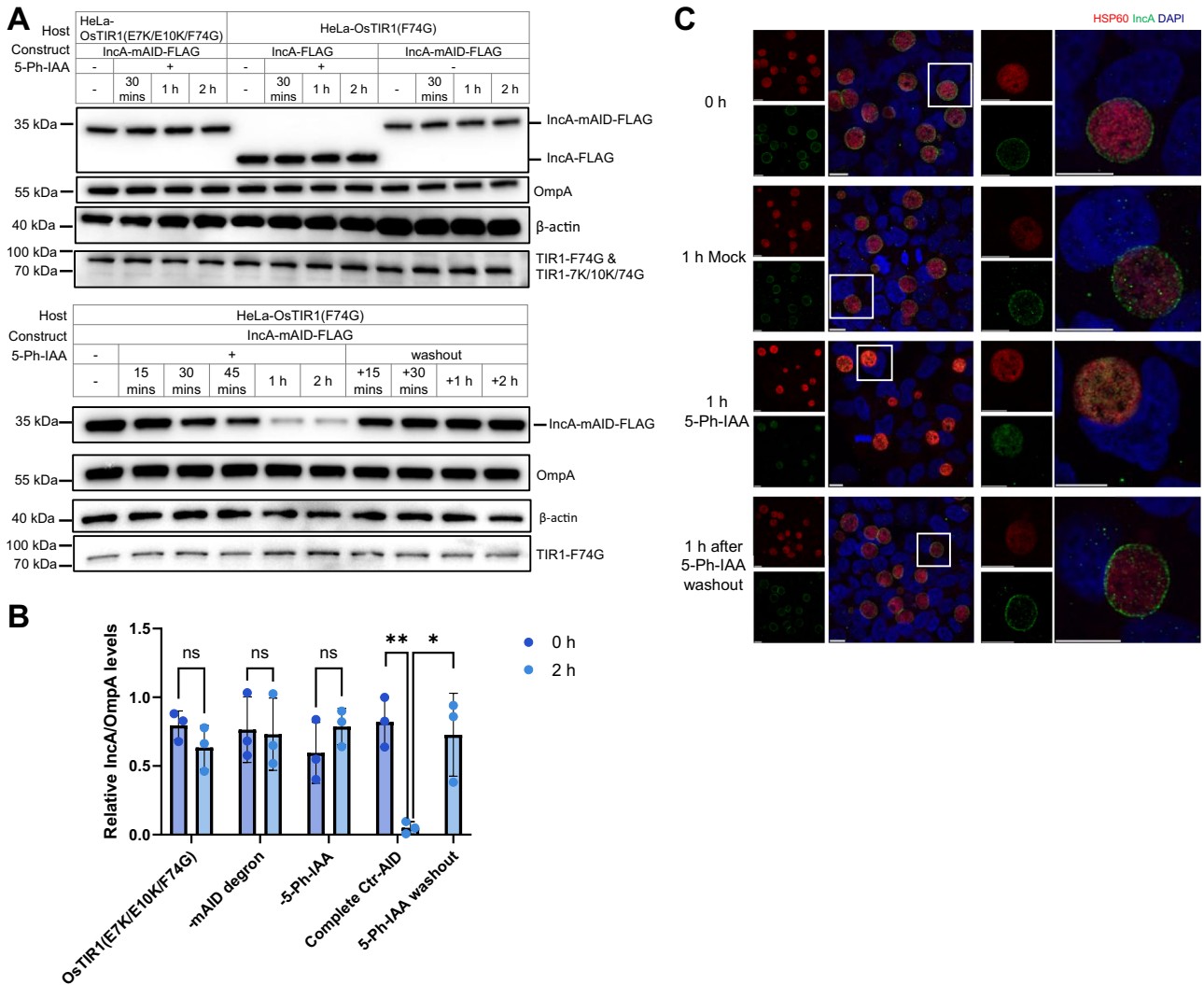

**Fig. 5 | Ctr-AIDE system enables conditional control of IncA expression in *Chlamydia*. A** Ctr-AIDE-mediated regulation of IncA-mAID expression. Degradation and re-expression protocols were performed as described for Cdu1. IncA (anti-FLAG) and OsTIR1 mutants (TIR1-F74G denotes OsTIR1(F74G)-9×Myc, and TIR1-7K/10 K/74 G denotes OsTIR1(E7K/E10K/F74G)-9×Myc; anti-Myc) levels were monitored, with OmpA as a loading control and β-actin as a host cell control. **B** Quantification of IncA degradation. FLAG levels normalized to OmpA (mean ± SD) from three independent immunoblot replicates (one shown in **A**). Significance assessed by two-tailed paired t-test (\*\**p* < 0.01, \**p* < 0.05; n.s., not significant, exact *p* values were provided in Supplementary Data 3). **C** Immunofluorescence microscopy of IncA depletion dynamics. IncA (green, anti-FLAG), inclusions (Hsp60, red), and DNA (DAPI, blue). The white square in the left panel marks the region enlarged in the right panel. Scale bar = 10 μm. All experiments were repeated ≥3 times with consistent results. Source data are provided as a Source Data file.

Addition of 5-Ph-IAA induced rapid degradation of IncA-mAID within 1 hour, followed by re-expression within 30 min after 5-Ph-IAA washout. In addition, degradation required a functional E3 ligase OsTIR1(F74G), the mAID degron and addition of 5-Ph-IAA, since controls lacking any component or with dysfunctional E3 ligase retained stable IncA levels (Fig. 5A, B). Immunofluorescence microscopy also demonstrated the loss of inclusion membrane-localized IncA 1 hour after addition of 5-Ph-IAA and its restoration after 5-Ph-IAA washout (Fig. 5C).

### Assessing IncA function dynamics

We next evaluated the functional consequences of IncA depletion. To remain consistent with previous study of IncA function[48], following experiments were performed at MOI = 1. Consistent with previous reports[39,40,48], IncA degradation by addition of 5-Ph-IAA from 0-24 hpi yielded a subset of infected cells containing multiple inclusions (Fig. 6A). This is consistent with the IncA's essential role in mediating homotypic inclusion fusion.

Despite intensive research on the mechanism of inclusion fusion[40,48,49], it remains unclear whether IncA activity is continuously required to maintain fused inclusions or becomes dispensable once fusion is complete. Addressing this question has been challenging due to the long stability of membrane-integrated IncA, which limits the effectiveness of conventional silencing approaches. Continuous sRNA-mediated silencing of IncA expression[50] from 0 to 24 hpi effectively reduced IncA levels (Supplementary Fig. 13A), whereas silencing initiated at 24 hpi failed to deplete the protein, which remained stable for at least another 24 h under these conditions (Supplementary Fig. 13B, D). In contrast, AIDE enabled rapid depletion of membrane-integrated IncA within 1 h at 24 hpi (Supplementary Fig. 13C, D), providing a powerful tool to directly assess the temporal requirements of IncA function. To determine whether IncA's function is static or dynamic, we implemented time-resolved degradation schemes. We compared two conditions: (i) Expression pause: IncA expressed from 0 to 24 hpi, then degraded until 40 hpi (Treatment 4). (ii) Rescue: IncA degraded from 0 to 24 hpi, then re-expressed via 5-Ph-IAA washout until 40 hpi

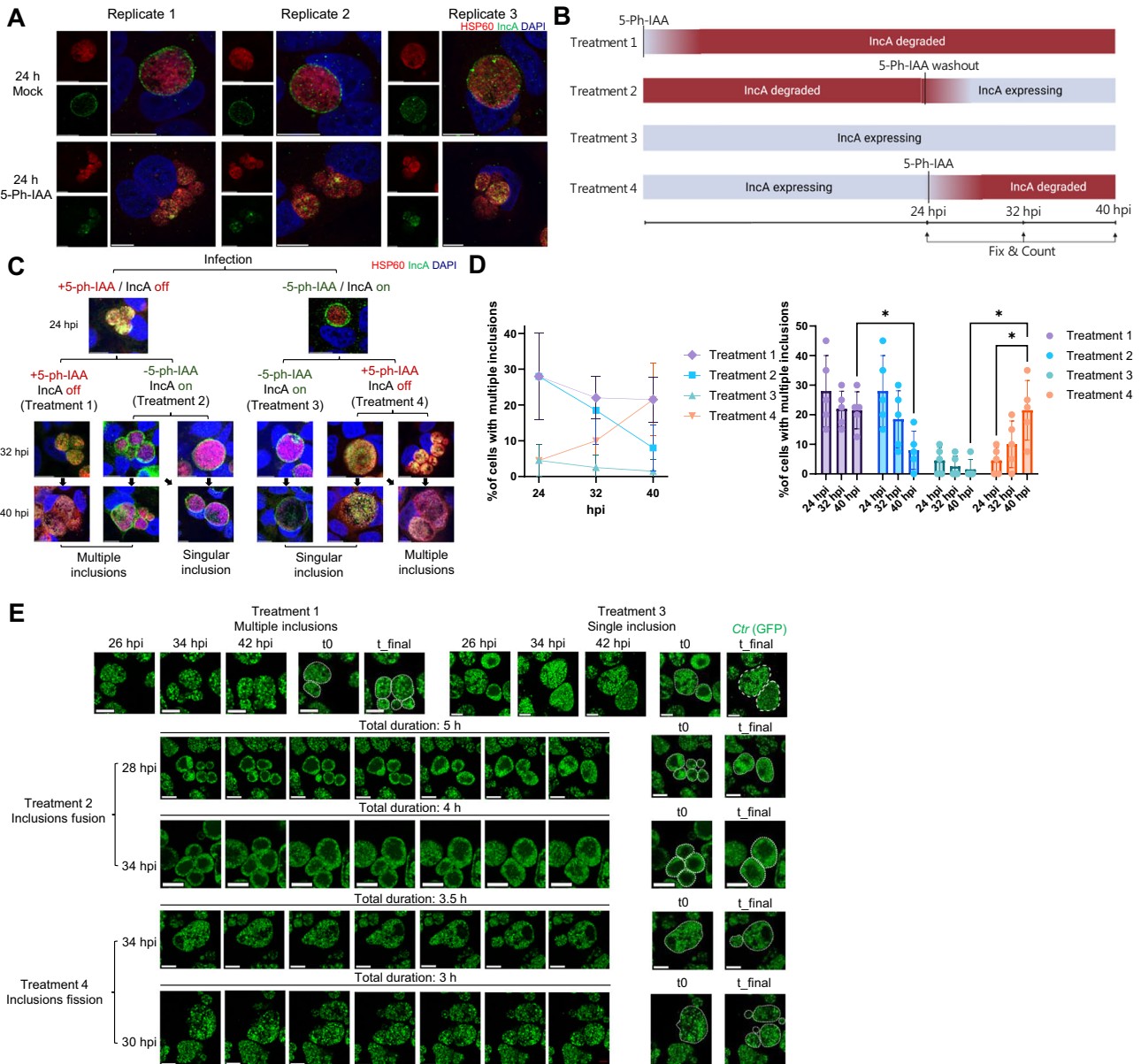

**Fig. 6 | IncA degradation reveals a continuous requirement for IncA in maintaining homotypic inclusion fusion during the *C. trachomatis* life cycle. A** IncA degradation induces a multi-inclusion phenotype. HeLa cells expressing OsTIR1(F74G) infected with IncA-mAID strains (MOI = 1) were treated with 5-Ph-IAA (IncA degraded) or DMSO (control) from infection to 24 hpi. Degraded samples show cells containing multiple inclusions, whereas controls form single inclusions. IncA (green, anti-FLAG), inclusions (Hsp60, red), and DNA (DAPI, blue). Scale bar = 10 μm. **B** Schematic of IncA termination and re-expression experiments. **C** Immunofluorescence microscopy of inclusion phenotypes (MOI = 1) under Treatment 1-4 at the indicated time points. **D** Quantification of host cells containing multiple inclusions (≥ 2) upon Treatment 1-4. Data represents mean percentage ± SD from 200 cells counted per condition across five independent biological replicates. Significance assessed by two-way ANOVA with Tukey Multiple comparisons test (*$p < 0.05$; not significant was not shown, exact $p$ values were provided in

Supplementary Data 3). **E** Live-cell imaging reveals inclusion fission and fusion upon IncA degradation and re-expression. Representative time-lapse images from Treatments 1-4 illustrate inclusion fission during IncA degradation (Treatment 4) and inclusion fusion following IncA re-expression (Treatment 2). Continuous IncA depletion (Treatment 1) resulted in persistent inclusion fission, whereas continuous IncA expression (Treatment 3) yielded single inclusions. For treatment 1 and 3, the indicated hpi above marks the time at which the image was captured. For treatment 2 and 4, the indicated hpi on the left marks the time at which imaging was started, and the total imaging duration is indicated above the images. For each series, t0 and t_final denote the first and last frames, and inclusions were marked by white dashed line for clarity. *Ctr* (green, GFP), Scale bar = 10 μm. All experiments were replicated ≥3 times with consistent results. Source data are provided as a Source Data file.

(Treatment 2). Controls included continuous IncA expression (0-40 hpi, Treatment 3) or degradation (0-40 hpi, Treatment 1) (Fig. 6B).

Control treatments established the expected inclusion phenotypes: continuous IncA expression was associated with predominantly single inclusions per infected cell (Treatment 3), whereas sustained IncA degradation resulted in ~25% of infected cells containing multiple inclusions from 24 to 40 hpi, likely representing the subset of cells in

which IncA-dependent homotypic fusion would normally occur (Treatment 1) (Fig. 6C, D). Quantitative analysis further revealed that initiating IncA degradation at 24 hpi ("Expression pause," Treatment 4) increased the fraction of infected cells containing ≥2 inclusions compared to the continuous-expression control (Fig. 6C, D). In contrast, restoring IncA after washout at 24 hpi ("Rescue," Treatment 2) reduced the proportion of cells with multiple inclusions toward control levels (Fig. 6C, D).

Because the penetrance of multi-inclusion phenotypes can depend on infectious input[51,52], we examined the effect of IncA depletion and re-expression across different MOIs. Using the same Treatment 1-4 scheme, we quantified the number of inclusions per infected cell at 24 and 40 hpi at MOI = 0.1, 1, and 10 (Supplementary Fig. 14). At MOI = 0.1, no significant differences in inclusion number were observed across treatments. In contrast, at MOI = 1 and MOI = 10, IncA depletion at 24 hpi (treatment 4) increased the number of inclusions per infected cell at 40 hpi, whereas IncA re-expression at 24 hpi (treatment 2) shifted this distribution back toward lower inclusion numbers at 40 hpi (Supplementary Fig. 14). These results are consistent with previous studies showing that IncA-related multiple-inclusion phenotypes become more apparent at higher MOI[51,52].

To determine whether these changes in inclusion number per cell reflect underlying fusion and fission dynamics, we performed live-cell imaging under the same treatment conditions at MOI = 1. Live-cell imaging directly captured these dynamics, showing inclusion fission upon rapid IncA depletion started at 24 hpi and inclusion fusion following IncA re-expression started at 24 hpi (Fig. 6E, Supplementary Movies 1-12). These results provide direct mechanistic evidence that IncA is not only required for the establishment of homotypic inclusion fusion, but also for maintenance of the fused-inclusion state, as loss of IncA leads to inclusion fission.

To rule out that the multi-inclusion phenotype is an artifact of the AIDE degradation system rather than a consequence of IncA loss, we performed a control experiment in which Cdu1 was degraded continuously from infection through 48 hpi and inclusion morphology was assessed. In contrast to IncA depletion, prolonged Cdu1 degradation did not increase the fraction of infected cells containing multiple inclusions (Supplementary Fig. 15), indicating that multi-inclusion phenotype is not a general outcome of AIDE activation and OsTIR1(F74G) recruitment to the inclusion and supporting the conclusion that the phenotype is attributable to IncA depletion.

To align with the Cdu1 functional analyses, we also tested the effects of IncA manipulation on chlamydial metabolism, developmental progression, and progeny-associated secondary infection burden using the same treatment scheme shown in Fig. 6B.

In primary cells, IncA degradation triggered multi-inclusion phenotype (Supplementary Fig. 16A). In the 2D PMRT model, IncA depletion reduced chlamydial metabolic activity (Supplementary Fig. 16B, left, treatment 1, 4 vs treatment 3). Notably, re-expression of IncA from 24-40 hpi did not fully restore metabolic activity (Supplementary Fig. 16B, left, treatment 2 vs treatment 3), suggesting that sustained IncA activity is required to maintain normal metabolic fitness in primary cells. In contrast, IncA degradation in HeLa cells did not measurably affect metabolic activity (Supplementary Fig. 16B, right).

In primary cells, prolonged IncA depletion from early infection through 40 hpi (treatment 1) impaired developmental progression, as indicated by reduced OmcB accumulation (Supplementary Fig. 16C, D; treatment 1 vs treatment 3), and was accompanied by reduced progeny-associated secondary infection burden in reinfection assays (Supplementary Fig. 16E, F; treatment 1 vs treatment 3). Restricting IncA depletion to 24-40 hpi caused a modest, non-significant reduction in OmcB levels (Supplementary Fig. 16C, D; treatment 4 vs treatment 3), yet this was accompanied by a significant reduction in progeny-associated secondary infection burden (Supplementary Fig. 16E, F; treatment 4 vs treatment 3), suggesting that even a subtle impairment in late-stage EB formation can translate into a measurable reduction in progeny-associated secondary infection burden. Consistent with these findings, IncA re-expression at 24 hpi did not fully restore developmental progression and secondary infection burden, mirroring the incomplete recovery of inclusion fusion (Supplementary Fig. 16C–F; treatment 2 vs treatment 3). In HeLa cells, by contrast, IncA degradation did not significantly alter OmcB levels or progeny-associated secondary infection burden (Supplementary Fig. 17A–D).

Together, these results indicate that IncA is particularly important for productive infection in primary cells, where loss of IncA disrupts inclusion fusion and may increase susceptibility to host restriction mechanisms, whereas IncA is largely dispensable for chlamydial metabolic activity and developmental progression in HeLa cells.

## Discussion

Traditional genetic tools (e.g., classical knockout, transposon mutagenesis, silencing technologies) have proven inadequate for resolving the dynamic function of secreted proteins in pathogenic bacteria[53–57]. Conditional tools such as CRISPRi and engineered sRNAs allow inducible repression but are hindered by issues including incomplete knockdown, promoter leakiness, plasmid stability and expression constraints, off-target and polar effects, delayed phenotypes for stable proteins, and challenges in restoring target gene expression[32,50,54]. By repurposing the host ubiquitin-proteasome system, AIDE overcomes these limits, enabling rapid (minutes-wise), reversible depletion of bacterial effectors at the host-pathogen interface.

One of the most notable advantages of AIDE lies in its versatile application which circumvents the complex molecular design required to establish functional PROTACs for TPD, which is a prohibitive barrier for studying many secreted effector proteins whose structures remain uncharacterized[58,59]. Its minimalist design, featuring a small 7 kDa mAID degron fused to targets that need to be accessible to host cells cytosol, ensures seamless genomic integration and preserves native protein function, as demonstrated for Cdu1 and IncA. AIDE also bypasses several shortcomings of bacteria-specific strategies like BacPROTACs, which depend on poorly defined bacterial proteases (e.g., ClpCP)[60], and the delivery of compounds over two or more membranes. Unlike transcriptional or translational knockdown methods, AIDE operates post-translationally, avoiding off-target effects on adjacent genes[61,62].

AIDE uniquely enables causal analysis of secreted effector function at defined times, a capability lacking in conventional transcriptional or translational silencing methods such as CRISPRi or engineered sRNAs. Although these approaches can suppress expression when initiated early and allow re-expression after inducer washout, they rarely achieve rapid depletion of pre-synthesized effectors during the mid to late infection cycle (Supplementary Fig. 13). This limitation arises from the long half-lives of membrane-associated proteins and pathogen-encoded countermeasures such as the deubiquitinase Cdu1 and the fusogen IncA. As a result, two constraints emerge: (i) phenotypes are typically revealed only upon re-expression rather than acute loss, and (ii) prolonged early inhibition can drive bacterial adaptation and persistence, confounding interpretation of single-effector loss[63]. By contrast, AIDE affords minute-scale, reversible control, enabling immediate degradation and rapid re-expression across precisely defined infection windows, with the potential for repeated on/off toggling. These properties reveal effector dynamics previously inaccessible to silencing-based strategies, as demonstrated for Cdu1 and IncA.

We demonstrate that the ubiquitin-directed degradation of transmembrane effectors harnesses the host p97 (VCP) system to enable efficient, proteasome-dependent clearance. p97 facilitates this process by actively extracting ubiquitinated substrates from membranes and delivering them to the cytosolic proteasome for degradation[44]. This mechanism likely confers high efficiency in depleting otherwise stable, membrane-anchored proteins, an effect not achievable with conventional knockdown approaches. Consistent with this, we detected Cdu1-mAID ubiquitination as early as 5 min after 5-Ph-IAA addition (Supplementary Fig. 3), with protein levels reduced by ~70% just 15 min later (Fig. 1D). Similarly, Ctr-AIDE enabled near-complete depletion of the stably membrane-integrated effector IncA within 1 h (Fig. 5). Auxin washout allows cytoplasmic proteins to rapidly fold and be re-exported by the T3SS, restoring membrane

activity for near-instantaneous functional analysis, as demonstrated by Cdu1's suppression of autophagy signaling within 1 h. Importantly, Ctr-AIDE induced rapid, minute-scale ubiquitin conjugation and degradation even of the deubiquitinase Cdu1. This finding was unexpected, as (i) the auto-acetyltransferase activity of Cdu1 on lysine residues has been proposed to protect it from K48-linked ubiquitination[38], and (ii) its potent, broad K48-directed deubiquitinase activity would be expected to counteract K48-mediated proteasomal degradation. Thus, enforced SCF E3 complex engagement via Ctr-AIDE overrides these protective activities, validating the approach as a robust strategy to acutely eliminate even host defense-evasive effectors.

By applying Ctr-AIDE to *Chlamydia* Incs, we resolved the spatio-temporal dynamics of two critical effectors. Cdu1 functions as a dual-phase sentinel, coordinating its enzymatic activities to safeguard the inclusion niche. Intriguingly, autophagy marker accumulation was delayed until 4 hours post-Cdu1 degradation, with no significant increase observed within the first hour. We attribute this delay to Cdu1's acetyltransferase activity. Previous studies indicate that Cdu1-mediated acetylation protects distinctive Incs from host ubiquitination[38,64]. Consequently, during the initial hour post-degradation, these effectors remained acetylated and resistant to ubiquitination. Over time, however, host deacetylases (e.g., HDACs) progressively deacetylated Incs, rendering Incs susceptible to ubiquitin-mediated tagging[65,66]. In contrast, re-expression experiments revealed that only mature, inclusion membrane-localized Cdu1 clears autophagy markers within 1 h, demonstrating its rapid, compartment-specific activity. This highlights its role as a frontline defender against host immune tagging.

During mid-late development (16–48 hpi), Cdu1 transitions to sustaining nutrient acquisition, metabolic activity and development. Time-resolved degradation showed that Cdu1 is essential in this window, coinciding with inclusion expansion, nutrient scavenging, and RB-to-EB differentiation. Because only inclusion membrane-localized Cdu1 is degraded, and previous studies show that growth defects in Cdu1-deficient strains are not caused by autophagy-mediated clearance, we propose that Cdu1 degradation during this critical window impairs Golgi vesicle recruitment[37]. This, in turn, starves replicating RBs of lipids, nucleotides, and ATP required both for binary fission and for the energy-intensive conversion into infectious EBs. Consistent with this model, sustained Cdu1 depletion reduced progeny-associated secondary infection burden, whereas short-term depletion had no measurable effect. In summary, the Cdu1's instant autophagy clearance and sustained nutrient acquisition emphasize its flexibility and its important role in *Ctr* developmental progression.

Conversely, IncA maintains homotypic inclusion fusion through persistent activity of its SNARE-like domain (SLD1/2). Strikingly, following inclusion fusion, degradation of inclusion membrane-bound IncA triggered vacuole fission. In primary cells, IncA depletion also reduced chlamydial metabolic activity, impaired EB-associated features, and decreased progeny-associated secondary infection burden, linking loss of inclusion fusion, as reflected by the multi-inclusion phenotype, to compromised developmental output. This dynamic equilibrium demonstrates that homotypic inclusion fusion is not a set-and-forget state but an IncA-dependent steady state that demands continuous ICS-like IncA assemblies to preserve the fused-inclusion state, rather than acting solely during fusion initiation. By establishing the maintenance phase as both IncA-dependent and reversible, our data identify a vulnerable window in which disrupting IncA-SNARE interactions could separate mature inclusions, alter the inclusion niche and thereby reduce productive development.

These insights exemplify how Ctr-AIDE bridges temporal resolution with subcellular precision, unmasking effector roles in developmental timing and niche maintenance. The system's capacity to decouple sequential functions such as autophagy evasion, nutrient scavenging, and maintenance of homotypic inclusion fusion reveals

stage-specific therapeutic vulnerabilities. Targeting these phases could halt *Chlamydia*'s developmental cycle, offering a roadmap for precision antimicrobial design.

More broadly, AIDE harnesses eukaryotic machinery to control pathogen proteins, establishing a host-directed strategy for manipulating intracellular bacteria. While sophisticated protein-control systems are well established in human cells, they are difficult to implement in bacteria, particularly intracellular species with limited genetic tractability[67]. By contrast, host-directed strategies such as AIDE directly exploit host-encoded pathways to regulate pathogen proteins with temporal precision, independent of the pathogen's biology, and can potentially circumvent microbial defense systems.

At present, AIDE targets inclusion membrane proteins that are directly accessible from the host cytosol. In combination with inducible delivery systems, analogous approaches could be extended to effectors located within the bacterial cytosol[68–70]. Because this strategy leverages host ubiquitin–proteasome machinery, it should be broadly generalizable, not only to vacuole-resident obligate intracellular bacteria with similar lifestyles (e.g., *Coxiella burnetii*, *Ehrlichia/Anaplasma* spp.) but also to facultative intracellular pathogens that replicate within human cells, including *Salmonella enterica*, *Shigella* spp., *Legionella pneumophila*, *Brucella* spp., and *Mycobacterium tuberculosis*. We therefore envision AIDE as a modular, pathogen-tailored toolkit, with organism-specific variants such as Ctr-AIDE - forming an expandable arsenal for the precise manipulation of pathogen proteins and biology.

Conceptually, AIDE also shifts the balance of control from the pathogen to the host. Pathogenic infections are notoriously difficult to cure because evolved microbes co-opt host pathways for their own benefit. Tools such as AIDE establish a synthetic defensive system that reverses this hierarchy, enabling host cells to dictate pathogen function. With continued advances in TPD, it may soon become possible to regulate pathogen proteins without engineered degrons, by instead recruiting specific E3 ligases. This trajectory offers a promising framework for the development of next-generation anti-pathogen therapeutics.

In conclusion, AIDE advances the mechanistic dissection of host-pathogen interactions by merging precision degradation with temporal control. Its success in revealing the dynamic functions of *Chlamydia* effectors highlights the potential of host-directed tools to accelerate antimicrobial discovery targeting pathogen-secreted effector proteins. Ctr-AIDE therefore establishes a blueprint for studying effector dynamics in the very large group of pathogenic bacteria that secret effector proteins into host cells.

## Methods

### Ethical statement

All procedures involving animals were conducted in accordance with institutional and national guidelines and were approved by the Regierung von Unterfranken, Würzburg, Germany (license number: TV-AZ 55.2.2-2532-2-762). Experiments involving the human pathogen *C. trachomatis* were conducted in the Biosafety Level (BSL) 2 laboratory of the Chair of Microbiology at the University of Wuerzburg, which is registered with the Government of Lower Franconia under code 8791-1.30.

### Antibodies used

Antibodies and their working dilution are listed in Supplementary Table 4.

### Plasmids and *Escherichia coli* strains

All plasmids and primers are listed in Supplementary Tables 2 and Supplementary Data 2. The plasmids sequences are provided in the Supplementary Data 1. Primers were synthesized by Sigma-Aldrich, and the mAID degron sequence was synthesized by Integrated DNA

Technologies. Plasmid construction proceeded as follows: The *Cdu1* coding sequence (*CTL0247*) and its 250 bp promoter region were amplified from *Ctr* serovar L2/434/Bu genomic DNA. mAID degron and FLAG-tag sequences were fused to the C-terminus of Cdu1 via PCR using primers with 30 bp homology overlaps. For genomic knock-in, the pKW-L2 backbone[41] was modified by Gibson Assembly (New England Biolabs) to incorporate: (i) 3 kb homology arms flanking Cdu1, (ii) engineered Cdu1-mAID-FLAG or Cdu1-FLAG inserts, (iii) aadA-GFP selection cassette (spectinomycin resistance/GFP fusion).

Identical strategies were applied for IncA (*CTL0374*) knock-in constructs, with all assemblies verified by Sanger sequencing.

For pBOMB5-Tet-CtrR3-IncA_aadA plasmid, CtrR3 construct that inhibits IncA expression is introduced into pBOMB5 backbone using primers CtrR3-IncA_F and CtrR3-IncA_R.

*Escherichia coli* strains (Supplementary Table 3) were cultured in Lysogeny Broth (LB; Carl Roth) or on LB agar plates (1.5% w/v agar) at 37 °C. DH10B served as the host for plasmid propagation. Media were supplemented with ampicillin (100 μg/mL) or spectinomycin (50 μg/mL) as required. Glycerol stocks (50% v/v) of engineered strains were stored at -80 °C.

## Culture of organoids and organoid-derived 2D monolayers

C57BL/6 J mice (Mus musculus) were originally obtained from Charles River and subsequently bred and maintained in-house at the animal facility of the Biocenter, Julius-Maximilians-Universität Würzburg, Würzburg, Germany. Mice were housed in individually ventilated cages (IVCs) with a 12 h light/12 h dark cycle, at an ambient temperature of 23 °C ± 2 °C and relative humidity of 50-60%, with food and water provided ad libitum.

In this study, mice were used solely as a source of tissue for organoid generation. Female C57BL/6 J mice aged 8-9 weeks (*n* = 3) were used for isolation of female reproductive tract tissues for organoid culture. Only female mice were used, as the organoid model is based on the female reproductive tract. Mice were euthanized by cervical dislocation prior to tissue collection.

All procedures involving animals were conducted in accordance with institutional and national guidelines and were approved by the Regierung von Unterfranken, Würzburg, Germany (license number: TV-AZ 55.2.2-2532-2-762).

The entire reproductive tract (ovaries, oviducts, uterus and cervix) was carefully excised, washed twice in ice-cold PBS containing 1% penicillin/streptomycin, and any excess adipose or connective tissue was trimmed away.

Tissues were minced into approximately 1-2 mm pieces and digested in digestion buffer (PBS + 1 mg/mL collagenase I [Thermo Fisher] + Trypsin 5% [Gibco] + 50 μg/mL DNase I [New England Biolabs]) at 37 °C for 60 min with gentle rocking. After digestion, the cell/tissue slurry was passed through a 40 μm nylon strainer, washed once in PBS. Cells were collected by centrifugation (1000 × g, 5 min, 4 °C), then resuspended in Matrigel. Aliquots of 50 μL Matrigel domes were plated into pre-warmed 24-well plates and each well received 500 μL of organoid medium (30% Advanced DMEM/F12 [Thermo Fisher Scientific], 50% WNT CM [self-made], 10% R-Spondin CM [self-made], 10% Noggin CM [self-made], 1x B27 supplement [Thermo Fisher Scientific], 1.25 mM N-acetylcysteine [Sigma Aldrich], 50 ng/mL EGF [Peprotech], 100 ng/mL FGF-10 [Peprotech], 1 mM Nicotinamide [Sigma Aldrich], 0.5 uM TGF-β inhibitor [Tocris]). Cultures were maintained at 37 °C under 5% CO₂ in a humidified incubator.

Organoids were passaged every 7-14 days at splitting ratios ranging from 1:2 to 1:10, based on growth density, with medium replenished every 2-3 days. For organoid passage, briefly, Matrigel domes were dissolved in ice-cold organoid medium. Released organoids were collected by centrifugation (400 × g, 5 min, 4 °C) and digested in 1 mL of TrypLE Express (Thermo Fisher Scientific) for 15 min at 37 °C with intermittent pipette trituration to obtain small clusters. Digested cells

were pelleted again, resuspended in Matrigel, and plated in 24-well plates. Finally, 500 μL of fresh organoid medium was added to each well.

To generate 2D monolayers, organoids were released from Matrigel and dissociated into single cells using TrypLE at 37 °C for 10 min. Dissociated cells were seeded into CellAdhere™ Type I Collagen-pretreated microwell plates or flasks (STEMCELL Technologies) and maintained in organoid medium under identical conditions. Primary murine reproductive tract cells were not subject to STR authentication.

## Cancer cell lines and cultivation

All the cell lines are listed in Supplementary Table 1. HeLa 229, A-375, HCT 116, U-2 OS, and McCoy cells were maintained in RPMI-1640 medium (Gibco) supplemented with 10% fetal calf serum (FCS; Paa Laboratories), while HEK293T cells were cultured in DMEM (Dulbecco's Modified Eagle Medium; Gibco) with 10% FCS. All cells were incubated at 37 °C under 5% CO₂ humidified conditions. Human cell lines were authenticated by STR profiling and routinely tested for mycoplasma contamination.

## Lentivirus generation and transduction

Lentivirus constructs for stable OsTIR1(F74G) or OsTIR1(E7K/E10K/F74G) expression were produced in HEK293T cells co-transfected with pRRL_OsTIR1F74G or pRRL_OsTIR1(E7K/E10K/F74G), and helper plasmids psPAX2/pMD2.G using Lipofectamine 3000 (Thermo Fisher). Cancer cell lines were transduced with viral supernatants containing 5 μg/mL Polybrene (Merck Millipore). Organoids were transduced as previously published[71]. Briefly, the viral pellet was resuspended in 500 μl organoid medium containing 10 mM nicotinamide and 8 μg/ml polybrene. For transduction, organoids were dissociated with trypsin, mixed with 250 μl of the virus, spinoculated at 600 × g for 1 h at 37 °C, then incubated for 3 h at 37 °C. After washing, organoids were embedded in 20 μl Matrigel. Following two infection rounds, hygromycin selection (1 mg/mL; Carl Roth) was applied to organoid in sequential 24-hour PBS wash and medium renew cycles to eliminate untransduced cells.

## *Ctr* propagation and transformation

*Ctr* serovar L2 (434/Bu) was propagated in McCoy cells. For infections, host cells at 70% confluency were incubated with EBs at an MOI of 1 in RPMI-1640/10% FCS at 35 °C under 5% CO₂. Bacterial stocks were prepared by harvesting infected cells at 48 hpi via mechanical disruption with glass beads. Debris was pelleted by centrifugation (1000 × g, 10 min, 4 °C), and EBs were subsequently isolated from supernatants by ultracentrifugation (30,000 × g, 30 min, 4 °C). Purified EBs were resuspended in sucrose-phosphate-glutamic acid (SPG) buffer (75 g/L sucrose, 0.52 g/L KH₂PO₄, 1.22 g/L Na₂HPO₄, 0.72 g/L L-glutamic acid, pH 7.4) and stored at -80 °C.

Transformation was performed as described[72]. Briefly, 10 μg plasmid DNA and 1.6 × 10⁷ inclusion-forming units (IFUs) of EBs were mixed in CaCl₂ buffer (10 mM Tris-HCl pH 7.4, 50 mM CaCl₂), then incubated with 8 × 10⁶ McCoy cells for 20 min at room temperature. The mixture was transferred to T75 flasks containing McCoy's 5 A medium (Gibco) with 10% FCS. At 48 hpi, lysates were passaged onto fresh McCoy cells under selection with 500 μg/mL spectinomycin and 1 μg/mL cycloheximide (both Carl Roth), with passages repeated every 48 h. Recombinant strains were verified by fluorescence and immunoblotting (Supplementary Table 3).

## Degradation assays

OsTIR1(F74G)- or OsTIR1(E7K/E10K/F74G)-expressing cancer cell lines were seeded in 12-well plates and infected with specified *Chlamydia* strains at MOI = 1. Degradation was induced by adding 1 μM 5-Ph-IAA (5-phenylindole-3-acetic acid; MedChemExpress), 200 ng/mL

anhydrotetracycline (aTC; Merck Millipore) or DMSO control to the culture medium. To terminate degradation, cells were washed once with DPBS and replenished with fresh drug-free medium. For primary cell lines, identical protocols were applied except for higher infection loads (MOI = 10), except for stating otherwise.

All samples including Mock (untreated controls) were harvested at 28 hpi for further analysis, with degradation and washout timed to synchronize endpoints (Fig. 2A). For multi-phase degradation-recovery experiments, treatments were applied in defined windows prior to harvest.

### Infectivity and metabolic assays
HeLa cells and primary PMRT cells were seeded in 12-well plates and infected with specified *Ctr* strains at MOI = 1. For metabolic assessment at 48 hpi, cells were lysed with 0.5 mm sterile glass beads in fresh HBSS; lysates were centrifuged (1000 × g, 10 min, 4 °C) and supernatants aliquoted for RT-qPCR (*gapA* quantification) and OmcB immunoblotting. For progeny-associated infection burden assays, parallel 48 hpi lysates were normalized by OmpA immunoblotting to equalize bacterial loads by adding fresh HBSS. Normalized supernatants were used to infect fresh HeLa monolayers, with progeny harvested at 24 hpi for immunoblot analysis.

### Inhibitor and IFN-γ treatments
Proteasome inhibitors (MG-132 at 10 μM, bortezomib at 1 μM; Cell Signaling), p97 inhibitor CB-5083 (10 μM; MedChemExpress), and ubiquitin-activating enzyme inhibitor TAK-243 (1 μM; Cell Signaling) were prepared as 1000× stock solutions in DMSO. Inhibitors were diluted to working concentrations in culture medium and applied 3 h prior to 5-Ph-IAA treatment, and kept in medium during the 1 h 5-Ph-IAA treatment. For IFN-γ induction, cells were pretreated with 50 U/mL recombinant human IFN-γ (Merck Millipore) for 2 h before infection.

### Indirect immunofluorescence
Immunofluorescence was performed on cells grown on glass coverslips. Following treatments or infections, cells were fixed with 4% paraformaldehyde (PFA; Carl Roth) for 15 minutes at room temperature, permeabilized with 0.2% Triton X-100 in PBS (30 min), and blocked in PBS containing 2% FCS (1 h). Primary antibodies diluted in blocking buffer were applied for 1 h at room temperature. After three 5-minutes PBS washes, secondary antibodies were incubated for 1 h protected from light. Coverslips were mounted using Fluoroshield mounting medium (Abcam) and imaged on a Leica STELLARIS 5 confocal microscope (63×/1.4 NA oil objective, inverted configuration). Detailed acquisition parameters, including measured image resolution, are provided with the raw data files. Images were processed using Leica *LAS X* software (3.10.0).

### Post-immunostaining gel embedding and expansion
Immunostained samples were post-fixed in 0.25% glutaraldehyde (10 min; Carl Roth). Specimens were embedded in monomer solution (1.375 M sodium acrylate, 2.5% acrylamide, 0.15% bis-acrylamide, 2 M NaCl, 1× PBS pH 7.4, 0.2% APS, 0.2% TEMED; all Sigma-Aldrich) and polymerized (1 h). Gels were digested in proteinase K buffer (8 U/mL in 50 mM Tris pH 8.0, 1 mM EDTA, 0.5% Triton X-100, 0.8 M guanidine HCl; 37 °C, overnight), then expanded in deionized water until saturation (4-5× linear expansion). Expanded gels were imaged using a Leica STELLARIS 5 confocal microscope (63×/1.4 NA oil objective, inverted configuration). Images were processed using Leica *LAS X* software (3.10.0).

### Live-cell time-lapse imaging
HeLa cells (3×10$^4$) were seeded in μ-Slide 8 Well (ibidi) plates 24 h before infection with the indicated *Chlamydia* strains at MOI 1. At 24 hpi, cells were treated as indicated and applied to imaging. Images

were acquired on a Leica STELLARIS 5 confocal microscope (40×/1.4 NA oil objective, inverted configuration) equipped with a Okolab bold line temperature and CO$_2$-controlled chamber. Imagines were captured using the resonant scanner every 10 min beginning at 26 hpi until 48 hpi. Images were processed and assembled into movies using Leica *LAS X* software (3.10.0).

### SDS-PAGE and immunoblotting
Cell lysates were prepared in RIPA Lysis and Extraction Buffer (Thermo Fisher Scientific) supplemented with Protease and Phosphatase Inhibitor Cocktail (Thermo Fisher Scientific). Lysates were clarified by centrifugation (16,000 × g, 10 min, 4 °C), and supernatants were mixed with ROTI®Load 1 loading buffer (Carl Roth) followed by denaturation at 94 °C for 5 minutes. Proteins were resolved on 12% SDS-PAGE gels and electrophoretically transferred to nitrocellulose membranes using a Bio-Rad Trans-Blot Turbo system (7 V, 25 min). Membranes were blocked with EveryBlot Blocking Buffer (Bio-Rad) for 10 min at room temperature, followed by overnight incubation with HRP-conjugated primary antibodies diluted per manufacturer specifications. After three 5-minute washes with TBS-T (0.1% Tween-20), blots were developed with Immobilon Forte Western HRP Substrate (Merck Millipore) for 1 minute and imaged using an Intas Chem HR 16-3200 system (Auto exposures) or an iBright FL1500 Imaging System (Thermo Fisher Scientific). Protein band intensities were quantified by densitometry using ImageJ (v1.53) with background subtraction.

### Quantitative real-time PCR (qRT-PCR)
Total RNA was extracted using the RNeasy Mini Kit (Qiagen). cDNA was synthesized from 1 μg RNA using the High-Capacity cDNA Reverse Transcription Kit (Thermo Fisher Scientific) in 20 μL reactions (25 °C for 10 min, 37 °C for 120 min, 85 °C for 5 min). qPCR was performed with SYBR Green Master Mix (Applied Biosystems) on a StepOnePlus™ system (Applied Biosystems) under standard cycling conditions: 95 °C for 10 min, followed by 40 cycles of 95 °C for 15 s and 60 °C for 1 min. Primer sequences are in Supplementary Data 2. Relative expression was calculated by the 2 - ΔΔCt method with chlamydial *rrs* (16S rRNA) as the endogenous control.

### Statistics & reproducibility
No statistical method was used to predetermine sample size and the experiments were not randomized. No data were excluded from the analyses and the investigators were not blinded to allocation during experiments and outcome assessment. Statistical analyses were performed using GraphPad Prism v10.4.0 (GraphPad Software). Data are presented as mean ± SD from at least three independent biological replicates, unless otherwise specified in the figure legends. Normality and homoscedasticity assumptions were assessed for all parametric tests. Statistical significance ($\alpha = 0.05$) was determined using two-tailed Student's *t* tests for two-group comparisons or one-/two-way ANOVA with Dunnett's or Tukey's post hoc tests for multiple comparisons, as appropriate. Exact statistical tests and *p*-value thresholds are indicated in the corresponding figure legends and exact *p* values are provided in Supplementary Data 3. Figures were assembled in Microsoft PowerPoint using graphical outputs from GraphPad Prism. Biological schematics were created using Microsoft PowerPoint and BioRender.com.

### Reporting summary
Further information on research design is available in the Nature Portfolio Reporting Summary linked to this article.

## Data availability
Source data are provided with this paper. The Source Data file includes quantified western blot data, RT-qPCR values used for analysis, inclusion counts, and counts of p62-positive inclusions. Raw data files

exported directly from Leica *LAS X*, ImageJ, and the StepOnePlus system have been deposited in Figshare, and the corresponding DOIs are provided in the reference list[73–75]. All plasmids, cell lines, bacterial strains, and chemical reagents generated or used in this study are available from the corresponding author upon request and subject to a material transfer agreement (MTA). Source data are provided with this paper.

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

## Acknowledgements

We thank Paul Köhling for providing the IncA sRNA silencing construct.

## Author contributions

H.Z. and T.R. conceptualized the study. H.Z. designed and performed experiments and analyzed the data. Y.G., B.A., and N.D. performed experiments and E.W. provided materials and technical support. T.R. supervised the project. H.Z. and T.R. wrote the manuscript, E.W. and Y.G. reviewed and edited the manuscript. All authors approved the submitted version.

## Funding

This work was supported by grants from the German Research Foundation (DFG) SFB1583 (DECIDE) to T.R., and RTG 2243 to T.R., the German Cancer Aid (DKH: TACTIC) to E.W. and the European Research Council (ERC) ERC-2018-ADG/NCI-CAD to T.R. and ERC: PROTAC-PDAC-101087045 to E.W. The Deutsche Forschungsgemeinschaft (DFG) funded the Leica Stellaris 5 CLSM under project code INST93/1159-1 FUGG. Open Access funding enabled and organized by Projekt DEAL.

## Competing interests

The authors declare no competing interest.
