## [Transparent Peer Review file · Nature Communications]

Minute-scale control of ubiquitin-mediated degradation reveals dynamics of bacterial secreted effector-functions

Corresponding Author: Professor Thomas Rudel

Version 0:

Reviewer comments:

Reviewer #1

(Remarks to the Author)

In this report, Zhang et al. successfully apply an improved auxin-inducible degron (AID2) technology to control the levels of two Chlamydial effector proteins, Cdu1 and IncA, in cells infected with *Chlamydia trachomatis*. The authors refer to this approach as AIDE (auxin-inducible degradation of effectors), although the underlying system is essentially identical to AID2 (Yesbolatova et al., Nat. Commun. 2020).

A major advantage of this system is that degron-fused Cdu1 and IncA can be rapidly degraded and then re-expressed in a temporally controlled manner. The authors demonstrate this convincingly in Figures 1 and 2 in HeLa cells, and further in Figure 3 using primary cells that better mimic physiologically relevant conditions. In addition, they show that AIDE induces the degradation of Cdu1 in a p97-dependent manner (Fig. 2E), suggesting that the p97 segregase extracts membrane-anchored Cdu1 for subsequent proteasomal degradation. The authors also report that sustained Cdu1 depletion inhibits chlamydial growth, although this inhibitory effect is lost when Cdu1 is degraded after 16 hpi (Fig. 4). Subsequently, the authors manipulated IncA levels in Figures 5 and 6 and demonstrated that sustained IncA expression is required for both the formation and maintenance of the inclusion.

Overall, the experiments are carefully executed, and the presented data are clear and convincing. This methodology will be valuable for researchers studying bacterial infections and the functions of bacterial effector proteins in infected cells. One concern, however, is whether this approach should be designated as a new method (AIDE), given that it is conceptually identical to AID2.

I found that the following points should be clarified.

1. Figure S1

This is an important WB dataset showing the expression of degron-fused Cdu1 and IncA at the beginning of the study. However, Cdu1 and IncA were detected only using anti-FLAG antibody. Please add blot data using anti-Cdu1 and anti-IncA antibodies.

2. Cite figures appropriately

At several points, it is difficult to connect the main text to the corresponding figures because the figure citations are unclear or missing.

Some examples include:

- Lines 168–170: Which figures are referenced here? Possibly Figure 1C?
- Lines 187–189: Should refer to Figure 2C–E.
- Lines 270–271: Should refer to Figure 5A, B.
- Lines 287–291: Please double-check the figure citations.

3. Fig. 1B

The experimental scheme is difficult to follow. Could you present it in a manner similar to Figure S8?

4. Fig 1C, D

I understand that Figure 1D is a quantification of Figure 1C. However, the two appear quite different, especially regarding Cdu1 levels in OsTIR1(E7K/E10K/F74G) and complete Ctr-AIDE conditions. Similar data in primary cells (Fig. 3A, B) do not show this discrepancy. Why? I note that the OsTIR1(E7K/E10K/F74G) dataset shows large error bars. If necessary, the authors should repeat the experiments to confirm the consistency of the results.

5. Line 256

OmpA appears suddenly without explanation. Please briefly describe what OmpA is.

6. Fig. 4G-I

AIDE-mediated depletion of Cdu1 reduces Chlamydial growth and progeny production. However, it is difficult to judge the magnitude of this phenotype. Can Cdu1 Δ strains be included as a control?

7. Fig. 5

The authors showed that IncA can be degraded and re-expressed. Can they also assess the impact on Chlamydial growth and progeny production?

8. Plasmids and cell lines

Because this paper describes a methodology, it is important to state clearly whether the plasmids, cell lines, and other materials used in the study will be made available.

Reviewer #2

(Remarks to the Author)

This manuscript presents an innovative application of the auxin-inducible degron (AID) system to achieve rapid, conditional depletion of chlamydial effector proteins during infection. Adapting this approach to *Chlamydia trachomatis* is technically impressive and has the potential to be a powerful tool for dissecting effector function with temporal precision.

You first apply the system to Cdu1 and show that early degradation during infection in the PMRT cell model reduces chlamydial fitness and decreases production of infectious progeny. This is an interesting observation and is broadly consistent with prior work implicating Cdu1 in maintaining inclusion integrity and supporting the developmental cycle.

However, the manuscript appears to overinterpret these findings as evidence for a direct role in the developmental cycle. Because any perturbation that reduces chlamydial growth will necessarily impact EB production, it is difficult to distinguish specific developmental defects from general fitness effects based on the current data. A more cautious interpretation, or additional experiments aimed at separating these effects, would strengthen the conclusions.

You also apply the AID system to IncA, a well-characterized inclusion membrane protein required for homotypic inclusion fusion. Early degradation of IncA blocks inclusion fusion, as expected and consistent with the existing literature. However, when IncA degradation is induced later during infection, you report fragmentation and loss of inclusion integrity and interpret this as evidence that IncA is continuously required to maintain inclusion stability. This interpretation is difficult to reconcile with published studies showing that loss of IncA has no effect on inclusion integrity or the developmental cycle at infections performed at an MOI below 1, where homotypic fusion does not occur. In addition, naturally occurring incA mutants have been isolated from patients, suggesting that IncA is dispensable for infectivity in both culture and in vivo contexts.

An alternative explanation that warrants more thorough consideration is that recruitment of the AID machinery itself to the inclusion membrane perturbs inclusion integrity, either independently of or in combination with degradation of the target protein. If this is the case, the observed IncA phenotype—and potentially aspects of the Cdu1 phenotype—may reflect unintended consequences of targeting the AID system to the inclusion rather than the specific loss of the effector protein. Addressing this possibility is important for validating the approach. For example, targeting additional inclusion membrane proteins that are known to be nonessential would help determine whether inclusion disruption is a general consequence of AID-mediated degradation at the inclusion membrane.

Overall, the use of the AID system in *Chlamydia* is technically innovative and promising. However, additional controls and a more conservative interpretation of the phenotypes are needed to fully support the mechanistic conclusions and to clearly establish protein-specific effects versus system-level artifacts.

Reviewer #3

(Remarks to the Author)

In this manuscript, the authors describe the development of a system for temporal degradation of *C. trachomatis* effector proteins by leveraging the established auxin-inducible degron system (AID). The system relies on tagging the effector at its genomic locus with the mAID degron and infecting cells expressing OsTIR1 (F74G), which, in the presence of the auxin analog 5-Ph-IAA, ubiquitinates the mAID-tagged protein, triggering proteasomal degradation.

The authors validated the system using two *C. trachomatis* effector proteins, Cdu1 and IncA, which both localize to the inclusion membrane and have been partially characterized.

The manuscript is well written, flows well, and for the most part the figures are easy to navigate (see minor comment on Figure 6).

This is an elegant and powerful system, for which the authors provide compelling evidence that it results in specific, temporal, and reversible degradation of *C. trachomatis* inclusion membrane proteins. Another strength of the system is that it can be used in several cell lines, as well as in primary cells. As such, the system presented here will be of interest to

researchers investigating the role of effector proteins in *C. trachomatis* infection, as well as to researchers studying other intracellular pathogens. Thus, the novelty and impact of the technical aspects of this manuscript are very high.

Using the system, the authors provide compelling evidence for a role of Cdu1 in bacterial development (Figure 4). While some of the conclusions should be toned down (see minor comments), especially regarding a role in the production of infectious bacteria, overall the data support that the absence of Cdu1 early in the developmental cycle affects bacterial growth and differentiation. However, because controls to rule out effects of long-term treatment with 5-Ph-IAA or constant recruitment/attack by OsTIR1 (F74G) are not included, the authors should comment on the long-term effects of the system on development of the IncA-mAID strain. A 48 h treatment is presented in Figure 6B–C, but there is no mention of the effect on inclusion development beyond the fusion phenotype.

The major concern that distracts from the technical innovation and impact of the manuscript relates to the novel biological aspects put forward for IncA. In many instances, the authors refer to a role for IncA in inclusion stability (line 266, line 309), integrity (lines 296–297), fragmentation (line 303), fragmented inclusions (label of Figure 6C–D), and membrane destabilization (line 310). In the micrographs presented in Figure 6, none of the infected cells contain inclusions with compromised morphology; instead, infected cells contain either a single intact inclusion or multiple intact inclusions. Based on this, the data point toward inclusion fission in the absence of sustained IncA production. If true, this would be a novel observation that could provide insight into inclusion fusion dynamics. Further characterization of this phenotype would be required, for example via live imaging to monitor inclusion fusion and fission after 5-Ph-IAA washout.

Minor comments

- Line 120: *Chlamydia* should be italicized.
- Line 185: “without affecting untagged effectors.” This statement is based on a single effector (IncA). Other effectors were not tested. Suggest toning down the claim.
- Line 235: “autophagy signaling evasion (Fig. 4B)” is a strong statement when the assay is limited to p62 recruitment to the inclusion. While this implies evasion of autophagy, it does not prove it.
- Lines 236–237: “impairs *Chlamydia* growth in primary cells, but not in HeLa cells (Fig. S7).” Is Figure S7 intended to illustrate that Cdu1 is not required for growth in HeLa cells? If so, does gapA expression serve as a readout for growth, or for metabolic activity as in Figure 4C? This section is puzzling.
- Line 258: While OmpA is present in elementary bodies, its presence does not equate to infectivity. Only a standard IFU assay, in which infectious progeny are collected and used to infect fresh monolayers, can determine bacterial infectivity.
- Lines 290–291: Double-check labeling of the figure panels; panel C corresponds to sRNA initiation at 24 hpi.
- Lines 296 and 303: Figure 6B should be cited before Figure 6C.
- Figure 4B: The graph is confusing and difficult to read. The legend lists three conditions, yet only two bar graphs per time point are shown. Improving the color palette (rather than using similar shades of blue) and increasing the size of the legend symbols would improve readability.
- Figure 4C–D: Please define “NC.”
- Figure 6B–D: Consider improving readability and organization. In panel B, a logical order would be treatment 1 at the top and treatment 4 at the bottom, matching the left-to-right order in panel C and the order presented in the text.
- Figure 6C: Consider increasing IF contrast and signal intensity.
- Figure 6C: Correct spelling: “fragmented,” not “fragemented.”
- Figure 6D: Consider changing the color scheme; light and dark colors are difficult to distinguish.
- Figures (general): To improve readability, consider placing time points/treatments above the corresponding IF images rather than below them.

Version 1:

Reviewer comments:

Reviewer #1

(Remarks to the Author)

The authors have appropriately revised the manuscript, which is now almost ready for publication. The following two points should be addressed in the final version.

I could not find an explanation for the asterisk in the Cdu1 blot shown in Fig. 1A, 3A, and S1A. Please indicate its meaning in the figure legends. Is this band a degradation product of Cdu1-mAID-FLAG? If so, does it disappear when the cells are treated with 5-Ph-IAA?

I previously requested that the authors state the availability of the materials described in the manuscript, such as plasmids and cell lines. However, I could not find this information in the revised version. As noted earlier, since this is a methodology paper, it is important to clearly indicate how these materials will be made available to the research community.

Reviewer #2

(Remarks to the Author)

The authors have addressed the concerns about the *incA* targeting phenotype in the response to reviewers. In the response

to reviewers comments the authors add more detail and analysis of the *incA* targeting experiments and their data and conclusions seem to be more inline with the published data on *incA* knockouts, knockdowns and natural mutants. This assuages the concerns about potential off target effects of their system. However, the authors have not actually changed the wording of the manuscript. The authors mention that they have “Rather than referring to “fragmentation” or compromised “inclusion integrity,” we now focus specifically on homotypic inclusion fusion phenotypes.” However, the abstract still states that “the integral membrane fusogen *IncA* is continuously required for inclusion integrity” and this is repeated throughout the discussion with statements such as “*IncA*-dependent steady state that demands continuous ICS-like *IncA* assemblies to counteract host-driven membrane destabilization, not solely on *IncA*’s role in initiating fusion” and *IncA* is continuously required for inclusion integrity.” These statements are not supported by the updated data.

Reviewer #3

(Remarks to the Author)

I thank the authors for conducting additional experiments and addressing my previous comments and concerns. The revised manuscript is thereby improved.

A few points still require clarification regarding the role of *IncA*, which, as noted in the first round of review, represents an important and novel aspect of the manuscript.

- Statements indicating a role for *IncA* in “inclusion integrity” (lines 24, 451), “inclusion architecture” (line 445), “destabilizing the pathogen niche” (lines 456–457), and any other instances I may have missed should be removed. Additionally, the paragraph on lines 445–457 should be substantially revised to focus on the role of *IncA* in inclusion fusion/fission rather than inclusion integrity or stabilization.
- The live-imaging data presented in Figure 6 to monitor *IncA*-dependent inclusion fusion and fission are a welcome addition to the manuscript. However, it remains unclear what the authors intend the main take-home message of these observations to be. As noted above, both the Abstract and the Discussion emphasize inclusion stabilization, which is not directly supported by the data presented.
- In the description of the still images in Figure 6C–D (lines 308–316), it would be more appropriate to describe the observations in terms of single versus multiple inclusions per cell, as reflected in the quantification shown in Figure 6D. At this stage, it is premature to conclude whether fusion or fission events are responsible for the observed number of inclusions per cell.
- Lines 308–309: the authors state that “control treatments establish the expected inclusion phenotypes... sustained *IncA* degradation caused inclusion fission.” As noted above, the phenotype should instead be described as multiple inclusions per cell. More importantly, a multiple-inclusion phenotype is not expected under the conditions described. Infection with an *incA* mutant strain at an MOI of 1 does not result in multiple inclusions per cell; the inclusion fusion defect of *incA* mutants becomes apparent primarily at MOIs greater than 1. Similarly, the statement that “continuous *IncA* expression maintained fused inclusions” is not appropriate if the infection was initiated at an MOI of 1, where a single inclusion per cell would already be expected.

I am not questioning the validity of the observed results. However, I would encourage the authors to:

- Avoid overinterpretation of the data (i.e., distinguishing between single/multiple inclusions versus fusion/fission events).
- Use more precise language, as a fusion event can only occur when multiple inclusions are present, and this should be reconciled with the experimental design using an MOI of 1.
- Clarify the central take-home message of the live-imaging experiments and what these results imply regarding the role of *IncA*.

These points have been partially addressed in the authors’ response to Reviewer 2, where the authors provide several of the clarifications I was looking for. The graphs referenced in that response should be incorporated into the manuscript, and the cited studies should be used to discuss the data more thoroughly.

RESPONSE TO REVIEWER COMMENTS

We thank the reviewers for their excellent and constructive comments. Please find our response below.

Reviewer #1 (Remarks to the Author):

In this report, Zhang et al. successfully apply an improved auxin-inducible degron (AID2) technology to control the levels of two Chlamydial effector proteins, Cdu1 and IncA, in cells infected with *Chlamydia trachomatis*. The authors refer to this approach as AIDE (auxin-inducible degradation of effectors), although the underlying system is essentially identical to AID2 (Yesbolatova et al., Nat. Commun. 2020).

A major advantage of this system is that degron-fused Cdu1 and IncA can be rapidly degraded and then re-expressed in a temporally controlled manner. The authors demonstrate this convincingly in Figures 1 and 2 in HeLa cells, and further in Figure 3 using primary cells that better mimic physiologically relevant conditions. In addition, they show that AIDE induces the degradation of Cdu1 in a p97-dependent manner (Fig. 2E), suggesting that the p97 segregase extracts membrane-anchored Cdu1 for subsequent proteasomal degradation. The authors also report that sustained Cdu1 depletion inhibits chlamydial growth, although this inhibitory effect is lost when Cdu1 is degraded after 16 hpi (Fig. 4). Subsequently, the authors manipulated IncA levels in Figures 5 and 6 and demonstrated that sustained IncA expression is required for both the formation and maintenance of the inclusion.

Overall, the experiments are carefully executed, and the presented data are clear and convincing. This methodology will be valuable for researchers studying bacterial infections and the functions of bacterial effector proteins in infected cells. One concern, however, is whether this approach should be designated as a new method (AIDE), given that it is conceptually identical to AID2.

We thank the reviewer for the overall positive feedback.

I found that the following points should be clarified.

1. Figure S1

This is an important WB dataset showing the expression of degron-fused Cdu1 and IncA at the beginning of the study. However, Cdu1 and IncA were detected only

using anti-FLAG antibody. Please add blot data using anti-Cdu1 and anti-IncA antibodies.

We performed the immunoblot and detected Cdu1 and IncA with antibodies, as suggested by the reviewer. The new data were included in Fig. S1A and S1B and the legend was revised accordingly.

2. Cite figures appropriately

At several points, it is difficult to connect the main text to the corresponding figures because the figure citations are unclear or missing.

Some examples include:

- Lines 168–170: Which figures are referenced here? Possibly Figure 1C?

Revised, see line 172.

- Lines 187–189: Should refer to Figure 2C–E.

Revised, see line 195.

- Lines 270–271: Should refer to Figure 5A, B.

Revised, see line 283.

- Lines 287–291: Please double-check the figure citations.

Revised, see lines 308-316.

3. Fig. 1B

The experimental scheme is difficult to follow. Could you present it in a manner similar to Figure S8?

We revised the figure as suggested.

4. Fig 1C, D

I understand that Figure 1D is a quantification of Figure 1C. However, the two appear quite different, especially regarding Cdu1 levels in OsTIR1(E7K/E10K/F74G) and complete Ctr-AIDE conditions. Similar data in primary cells (Fig. 3A, B) do not show this discrepancy. Why? I note that the OsTIR1(E7K/E10K/F74G) dataset shows large error bars. If necessary, the authors should repeat the experiments to confirm the consistency of the results.

We thank the reviewer for pointing out this discrepancy. We agree that the original quantification in Fig. 1D (and similarly in Fig. 3B and Fig. 5B) appeared inconsistent

with the corresponding representative immunoblots, particularly for the OsTIR1(E7K/E10K/F74G) and complete Ctr-AIDE conditions and was associated with relatively large error bars.

Upon re-examining the raw images and the analysis by the iBright Analysis Software from Thermo Fisher, we found that automated ROI/band selection did not consistently capture the intended Cdu1 band across replicates in these panels. In several lanes, the software included adjacent nonspecific signal, which led to overestimation of Cdu1 or IncA intensity (~2x higher) and inflated variance. Importantly, this issue affected only Fig. 1D, Fig. 3B and Fig. 5A, when +-mAID samples are presented together; the underlying immunoblots and experimental replicates were unchanged.

We therefore re-quantified the datasets using ImageJ, with a consistent, manually defined ROI for the Cdu1 band in each lane with uniform background subtraction, applying identical analysis parameters across all conditions and replicates. Although the automated ROI/band-detection issue was identified only for these panels, for consistency and transparency we re-analyzed all immunoblot quantifications in the manuscript using the same manual ROI selection. This re-analysis reduced variability and brought the quantification into closer agreement with the representative blots while preserving the overall effect sizes (n=3) and trends. The corrected quantifications are now shown in the revised manuscript.

Following re-quantification, the statistical significance of several pairwise comparisons changed.

- (i) Fig. 1D: the comparison between Complete Ctr-AIDE at 15 min versus 30 min, previously not significant, now reaches $p < 0.05$.
- (ii) Fig. 2D: the comparison between DMSO-only and DMSO + 5-Ph-IAA, previously $p < 0.05$, is now $p < 0.0001$.
- (iii) Fig. 3B: the comparison between 5-Ph-IAA washout at 15 min versus 30 min decreased from $p < 0.01$ to $p < 0.05$.
- (iv) Fig. 5B: the comparison between Complete Ctr-AID 0 h versus 2h increased from $p < 0.05$ to $p < 0.01$, while the comparison between Complete Ctr-AID 2 h versus 5-Ph-IAA washout decreased from $p < 0.01$ to $p < 0.05$.
- (v) Fig. S5B: in A-375 cells, the comparison between 5-Ph-IAA at 0 h versus 2 h, previously not significant, now reaches $p < 0.05$. In addition, the comparisons of U-2 OS and A-375 relative to HeLa decreased from $p < 0.0001$ to $p < 0.01$.
- (vi) Fig. S12D: the comparison between AIDE 0 h and 24 h degradation increased from $p < 0.05$ to $p < 0.01$.

Importantly, the overall kinetics and directionality of Cdu1 and IncA degradation and re-expression remain unchanged, and the conclusions of the study are not affected.

To ensure transparency, we provide the uncropped blots and the underlying raw densitometry values for all replicates as source data in the revised supplementary materials.

5. Line 256

OmpA appears suddenly without explanation. Please briefly describe what OmpA is.

We included the following sentence to introduce OmpA: “OmpA (Major Outer Membrane Protein, MOMP) is highly expressed in EBs and is critical for providing structural integrity to the chlamydial outer membrane.” see lines 260-262.

6. Fig. 4G-I

AIDE-mediated depletion of Cdu1 reduces Chlamydial growth and progeny production. However, it is difficult to judge the magnitude of this phenotype. Can Cdu1Δ strains be included as a control?

We thank the reviewer for this suggestion. We included the Chlamydia Tn-Cdu1 strain, which has a transposon (Tn) insertion disrupting the *cdu1* gene. This strain has been extensively investigated and has been demonstrated to be affected in progeny formation in vitro and in vivo (<https://doi.org/10.1111/cmi.13136>; DOI: [10.7554/eLife.21465](https://doi.org/10.7554/eLife.21465)). We used this strain as a control and included the data in the experiments shown in Fig. 4C-H, Fig. S8 and in Fig. S10.

7. Fig. 5

The authors showed that IncA can be degraded and re-expressed. Can they also assess the impact on Chlamydial growth and progeny production?

We assessed the impact of IncA depletion on Chlamydia infection in HeLa and primary cells. We could not detect a major effect of IncA depletion on Chlamydia infection in HeLa cells, however, an infection -phenotype could be detected in organoid-derived primary cells. These data have been included in the revised version (see lines 329 – 353 and Fig. S14).

8. Plasmids and cell lines

Because this paper describes a methodology, it is important to state clearly whether the plasmids, cell lines, and other materials used in the study will be made available.

Materials will be made available upon request.

Reviewer #2 (Remarks to the Author):

This manuscript presents an innovative application of the auxin-inducible degron (AID) system to achieve rapid, conditional depletion of chlamydial effector proteins during infection. Adapting this approach to *Chlamydia trachomatis* is technically impressive and has the potential to be a powerful tool for dissecting effector function with temporal precision.

You first apply the system to Cdu1 and show that early degradation during infection in the PMRT cell model reduces chlamydial fitness and decreases production of infectious progeny. This is an interesting observation and is broadly consistent with prior work implicating Cdu1 in maintaining inclusion integrity and supporting the developmental cycle. However, the manuscript appears to overinterpret these findings as evidence for a direct role in the developmental cycle. Because any perturbation that reduces chlamydial growth will necessarily impact EB production, it is difficult to distinguish specific developmental defects from general fitness effects based on the current data. A more cautious interpretation, or additional experiments aimed at separating these effects, would strengthen the conclusions.

You also apply the AID system to IncA, a well-characterized inclusion membrane protein required for homotypic inclusion fusion. Early degradation of IncA blocks inclusion fusion, as expected and consistent with the existing literature. However, when IncA degradation is induced later during infection, you report fragmentation and loss of inclusion integrity and interpret this as evidence that IncA is continuously required to maintain inclusion stability. This interpretation is difficult to reconcile with published studies showing that loss of IncA has no effect on inclusion integrity or the developmental cycle at infections performed at an MOI below 1, where homotypic fusion does not occur. In addition, naturally occurring *incA* mutants have been isolated from patients, suggesting that IncA is dispensable for infectivity in both culture and in vivo contexts.

We thank the reviewer for highlighting this important point and for the opportunity to clarify our interpretation.

All IncA degradation and characterization experiments presented in Fig. 6 were performed at MOI = 1, and we now state this explicitly in the Fig. 6 legend and Methods to avoid ambiguity (lines 588, 642, 980 and 985).

To improve clarity and better align our descriptions with the established IncA literature, we have revised our terminology throughout the Results section. Rather than referring to “fragmentation” or compromised “inclusion integrity,” we now focus specifically on **homotypic inclusion fusion phenotypes**. In our immunofluorescence analyses, we did not observe ruptured or structurally “broken” inclusions. Instead, we observed an increased frequency of **multiple inclusions per**

cell, consistent with inclusion fission and/or failure of homotypic fusion. All corresponding text changes are highlighted in the revised manuscript (lines 310-314, lines 316-321, lines 446-448, lines 454-457, lines 981-982, Fig. 6).

Our findings are consistent with prior studies analyzing IncA function at similar infection conditions. For example, (<https://pmc.ncbi.nlm.nih.gov/articles/PMC4859576/>) primarily characterizes IncA phenotypes at MOI = 1, supporting that homotypic inclusion fusion events can still occur under these conditions. Consistent with this, in our continuous IncA depletion condition (Treatment 1), ~25% of infected cells contained multiple inclusions at 24 hpi. We interpret this as a population in which IncA-dependent fusion would normally occur; therefore, acute IncA degradation is expected to shift inclusion morphology toward the multiple-inclusion phenotype.

Importantly, fusion-related phenotypes have also been reported at lower MOI in experimental contexts that perturb IncA function acutely. For example, the anti-IncA microinjection study (performed at MOI ~0.1) reported disrupted inclusion fusion and inclusion fission following antibody-mediated inhibition of IncA (https://onlinelibrary.wiley.com/doi/epdf/10.1046/j.1462-5822.1999.00012.x?saml_referrer), indicating that fusion defects can still be revealed even when initial infectious input is low. Conversely, genetic IncA knockout studies at low MOI (e.g., MOI = 0.1) have reported predominantly single inclusions per cell (<https://journals.plos.org/plosone/article?id=10.1371/journal.pone.0083989>), underscoring that the penetrance of multi-inclusion phenotypes depends on MOI and experimental context.

To directly address the reviewer's concern, we have now repeated the Treatment 1–4 degradation paradigm across **MOI = 0.1, 1, and 10**, quantifying inclusion number per infected cell at 24 and 48 hpi. As anticipated:

- **MOI = 0.1:** No significant differences in inclusion number were observed across treatments.
- **MOI = 1 and 10:** We observed the same qualitative trends reported in the manuscript, namely, increased multi-inclusion phenotypes upon IncA depletion and restoration toward fused inclusions upon re-expression.

These new data are provided for the reviewer's evaluation (see below) and can be incorporated into the revised manuscript if desired.

Finally, to directly visualize the dynamics underlying these phenotypes, we performed live-cell imaging following Inca degradation and re-expression. These experiments captured inclusion fission and fusion events in real time and are now included in the revised manuscript (lines 316-321; Fig. 6E), with corresponding Supplementary Movies 1–12.

Together, these clarifications, revised terminology, additional MOI comparisons, and live-cell analyses reconcile our observations with the published literature and support a model in which Inca is continuously required for homotypic fusion and fusion-dependent inclusion morphology under conditions where such fusion events occur.

An alternative explanation that warrants more thorough consideration is that recruitment of the AID machinery itself to the inclusion membrane perturbs inclusion integrity, either independently of or in combination with degradation of the target protein. If this is the case, the observed Inca phenotype—and potentially aspects of the Cdu1 phenotype—may reflect unintended consequences of targeting the AID system to the inclusion rather than the specific loss of the effector protein.

Addressing this possibility is important for validating the approach. For example, targeting additional inclusion membrane proteins that are known to be nonessential would help determine whether inclusion disruption is a general consequence of AID-mediated degradation at the inclusion membrane.

We thank the reviewer for this comment. We had similarly considered the possibility that recruitment of the E3 ligase to the inclusion membrane could produce adverse phenotypes due to off-target effects.

To address this concern, we deliberately selected two secreted effectors with well-defined and mechanistically distinct functions - Cdu1 and Inca - for the present study. Notably, depletion of these proteins yields clearly different phenotypes: Cdu1 loss results in inclusion ubiquitination and p62 recruitment, whereas Inca depletion

leads to impaired homotypic inclusion fusion. The observation that each target produces a distinct and functionally consistent phenotype provides an internal cross-validation control. If E3 ligase recruitment were causing nonspecific off-target effects at the inclusion, we would expect overlapping or shared phenotypes across targets. Instead, the divergent outcomes support the conclusion that the observed effects arise from specific depletion of the respective effector proteins rather than from off-target consequences of the degradation system.

In addition, to exclude the possibility that the AIDE system or prolonged auxin exposure per se induces a multi-inclusion phenotype, we performed a control experiment in which Cdu1 was continuously degraded by 5-Ph-IAA treatment for 48 hours. We conducted immunofluorescence imaging and quantified the percentage of infected cells containing multiple inclusions.

In contrast to IncA depletion, prolonged Cdu1 degradation did not increase the proportion of cells with multiple inclusions (Fig. S13; lines 322-328). These results support the conclusion that the inclusion fission/multi-inclusion phenotype is not a general artifact of the degradation system but is specifically associated with IncA loss.

Overall, the use of the AID system in *Chlamydia* is technically innovative and promising. However, additional controls and a more conservative interpretation of the phenotypes are needed to fully support the mechanistic conclusions and to clearly establish protein-specific effects versus system-level artifacts.

We thank the reviewer for the overall positive feedback and the suggestion to include additional controls which we performed during the revision of the manuscript

Reviewer #3 (Remarks to the Author):

In this manuscript, the authors describe the development of a system for temporal degradation of *C. trachomatis* effector proteins by leveraging the established auxin-inducible degron system (AID). The system relies on tagging the effector at its genomic locus with the mAID degron and infecting cells expressing OsTIR1 (F74G), which, in the presence of the auxin analog 5-Ph-IAA, ubiquitinates the mAID-tagged protein, triggering proteasomal degradation.

The authors validated the system using two *C. trachomatis* effector proteins, Cdu1 and IncA, which both localize to the inclusion membrane and have been partially characterized.

The manuscript is well written, flows well, and for the most part the figures are easy to navigate (see minor comment on Figure 6).

This is an elegant and powerful system, for which the authors provide compelling evidence that it results in specific, temporal, and reversible degradation of *C. trachomatis* inclusion membrane proteins. Another strength of the system is that it can be used in several cell lines, as well as in primary cells. As such, the system presented here will be of interest to researchers investigating the role of effector proteins in *C. trachomatis* infection, as well as to researchers studying other intracellular pathogens. Thus, the novelty and impact of the technical aspects of this manuscript are very high.

Using the system, the authors provide compelling evidence for a role of Cdu1 in bacterial development (Figure 4). While some of the conclusions should be toned down (see minor comments), especially regarding a role in the production of infectious bacteria, overall the data support that the absence of Cdu1 early in the developmental cycle affects bacterial growth and differentiation. However, because controls to rule out effects of long-term treatment with 5-Ph-IAA or constant recruitment/attack by OsTIR1 (F74G) are not included, the authors should comment on the long-term effects of the system on development of the IncA-mAID strain. A 48 h treatment is presented in Figure 6B–C, but there is no mention of the effect on inclusion development beyond the fusion phenotype.

We assessed the impact of IncA depletion on *Chlamydia* infection in both HeLa cells and organoid-derived primary cells. In HeLa cells, we did not detect a major effect of IncA depletion on infection. In contrast, a significant infection phenotype was observed in organoid-derived primary cells. These data have been incorporated into the revised manuscript (lines 329 – 353; Fig. S14).

We share the reviewer's concern regarding potential long-term effects of OsTIR1(F74G) recruitment to inclusions and therefore incorporated a 48 h time point into our experimental design. IncA knockout strains have not been reported to exhibit a detectable phenotype in HeLa cells, consistent with our observations (Fig. S14). Thus, any adverse effects arising from prolonged OsTIR1(F74G) recruitment, beyond the expected inclusion fission/fusion phenotypes, should have been apparent in this system.

Because *C. trachomatis* L2 completes its developmental cycle in ~48 h in HeLa cells, nonspecific or off-target effects (i.e., independent of IncA degradation) that compromised inclusion integrity would be expected to impair bacterial infectivity. However, this was not observed (Fig. S14). Even continuous 48 h treatment with 5-Ph-IAA had no measurable impact on chlamydial infectivity in HeLa cells, despite efficient degradation of IncA at the inclusion membrane mediated by OsTIR1(F74G).

In addition, we included the Tn-cdu1 strain and the IncA knockout strain as loss-of-function controls for experiments assessing Cdu1 and IncA function, respectively

(Fig. 4C, F, H; Fig. S14B-F). Under prolonged 5-Ph-IAA treatment (48 h) to induce Cdu1 or IncA degradation, the resulting effects on chlamydial metabolic activity, developmental progression, and progeny-associated secondary infection burden closely phenocopied the respective genetic controls. These findings support the conclusion that the observed developmental phenotypes arise from specific target protein depletion rather than nonspecific consequences of extended 5-Ph-IAA exposure or OsTIR1(F74G) recruitment.

To further address Reviewer 2's concern that prolonged OsTIR1(F74G) activity might itself influence inclusion fission/fusion dynamics, we performed an additional control experiment in which Cdu1 was continuously degraded for 48 h using 5-Ph-IAA. Immunofluorescence imaging followed by quantification of infected cells containing multiple inclusions revealed that, in contrast to IncA depletion, prolonged Cdu1 degradation did not increase the proportion of cells with multiple inclusions (Fig. S13; lines 322-328).

Together, these results demonstrate that the inclusion fission/multi-inclusion phenotype is not a general artifact of the degradation system but is specifically associated with IncA loss. Overall, our data support the specificity of the AID system for its tagged targets under the conditions used in this study.

The major concern that distracts from the technical innovation and impact of the manuscript relates to the novel biological aspects put forward for IncA. In many instances, the authors refer to a role for IncA in inclusion stability (line 266, line 309), integrity (lines 296–297), fragmentation (line 303), fragmented inclusions (label of Figure 6C–D), and membrane destabilization (line 310). In the micrographs presented in Figure 6, none of the infected cells contain inclusions with compromised morphology; instead, infected cells contain either a single intact inclusion or multiple intact inclusions. Based on this, the data point toward inclusion fission in the absence of sustained IncA production. If true, this would be a novel observation that could provide insight into inclusion fusion dynamics. Further characterization of this phenotype would be required, for example via live imaging to monitor inclusion fusion and fission after 5-Ph-IAA washout.

We agree with the reviewer that the manuscript requires revision to clarify the phenotype we see upon IncA depletion. As already explained as response to the comment of reviewer 2, we have revised our terminology throughout the Results section to focus on homotypic inclusion fusion, rather than “fragmentation” or compromised “inclusion integrity.” In our immunofluorescence analyses we did not observe ruptured or “broken” inclusions; instead, we observed an increased frequency of multiple inclusions per cell, consistent with inclusion fission and/or failure of homotypic fusion. All corresponding text changes in the revised manuscript are highlighted in red (lines 310-314, lines 316-321, lines 446-448, lines 454-457, lines 981-982, Fig. 6).

We followed the advice of the reviewer and performed live cell imaging to monitor inclusion fission upon IncA depletion. We indeed see inclusion fission in the absence of IncA and fusion, if we wash out 5-Ph-IAA. We included these experiments and data in the revised manuscript, lines 316 to 321, in Fig 6E and as Supplementary Movies 1-12.

Minor comments

- Line 120: Chlamydia should be italicized.

Done, see line 122.

- Line 185: “without affecting untagged effectors.” This statement is based on a single effector (IncA). Other effectors were not tested. Suggest toning down the claim.

‘Without affecting untagged effectors’ was deleted, see line 187.

- Line 235: “autophagy signaling evasion (Fig. 4B)” is a strong statement when the assay is limited to p62 recruitment to the inclusion. While this implies evasion of autophagy, it does not prove it.

We changed the sentence to “..., while re-expression led to loss of p62 staining”, see line 237.

- Lines 236–237: “impairs Chlamydia growth in primary cells, but not in HeLa cells (Fig. S7).” Is Figure S7 intended to illustrate that Cdu1 is not required for growth in HeLa cells? If so, does gapA expression serve as a readout for growth, or for metabolic activity as in Figure 4C? This section is puzzling.

We thank the reviewer for pointing out this source of confusion. Indeed, Fig. S7 is intended to demonstrate that Cdu1 depletion does not measurably reduce chlamydial metabolic activity in HeLa cells under our assay conditions.

In both Fig. S7 and Fig. 4C, gapA expression is used as a readout of metabolic activity rather than as a direct measure of bacterial growth or infectivity. While metabolic activity is generally expected to correlate with bacterial fitness and replication capacity, we agree that it should not be interpreted as a standalone metric of growth.

To clarify this point and avoid misinterpretation, we have revised the text to explicitly define gapA as a marker of metabolic activity (and, by extension, bacterial fitness). In addition, we repositioned the statement regarding the HeLa cell phenotype to the end of the paragraph to improve the logical flow of the argument (lines 238-250).

- Line 258: While OmpA is present in elementary bodies, its presence does not equate to infectivity. Only a standard IFU assay, in which infectious progeny are collected and used to infect fresh monolayers, can determine bacterial infectivity.

We agree with the reviewer that OmpA abundance does not directly equate to bacterial infectivity and that inclusion-forming unit (IFU) assays represent the gold standard for quantifying infectious progeny. We have therefore revised the manuscript to explicitly avoid claims of infectivity and to describe OmpA levels in secondary infections as a surrogate readout of progeny-associated secondary infection burden rather than infectious unit formation.

We emphasize that our conclusions are limited to effects on chlamydial developmental progression and progeny-associated secondary infection capacity, and we do not claim direct measurement of infectivity.

We have clarified these limitations in the Results, Discussion, and figure legends. I have re-written the relevant results, figure legend and discussion section. Lines 258-270 & 440-442 & 954-960.

- Lines 290–291: Double-check labeling of the figure panels; panel C corresponds to sRNA initiation at 24 hpi.

We have changed the labelling in Fig. S12 accordingly.

- Lines 296 and 303: Figure 6B should be cited before Figure 6C.

We have changed the order of presentation as suggested by the reviewer (Lines 302-311).

- Figure 4B: The graph is confusing and difficult to read. The legend lists three conditions, yet only two bar graphs per time point are shown. Improving the color palette (rather than using similar shades of blue) and increasing the size of the legend symbols would improve readability.

We have changed the color palette to green, red and blue, and increased the symbols size.

- Figure 4C–D: Please define “NC.”

We have changed the relevant labels in the figure from NC to DMSO.

- Figure 6B–D: Consider improving readability and organization. In panel B, a logical order would be treatment 1 at the top and treatment 4 at the bottom, matching the left-to-right order in panel C and the order presented in the text.

We have changed the order as suggested by the reviewer.

- Figure 6C: Consider increasing IF contrast and signal intensity.

Done.

- Figure 6C: Correct spelling: “fragmented,” not “fragemented.”

We have changed the phrase ‘fragmented inclusion’ into ‘multiple inclusions’

- Figure 6D: Consider changing the color scheme; light and dark colors are difficult to distinguish.

We have changed the color to a more contrast group.

- Figures (general): To improve readability, consider placing time points/treatments above the corresponding IF images rather than below them.

We changed the labelling of the IF images as suggested by the reviewer and placed the labelling of time points and treatments above IF images.

REVIEWER COMMENTS

Reviewer #1 (Remarks to the Author):

The authors have appropriately revised the manuscript, which is now almost ready for publication. The following two points should be addressed in the final version.

I could not find an explanation for the asterisk in the Cdu1 blot shown in Fig. 1A, 3A, and S1A. Please indicate its meaning in the figure legends. Is this band a degradation product of Cdu1-mAID-FLAG? If so, does it disappear when the cells are treated with 5-Ph-IAA?

We believe these represent putative truncated or degradation fragments of the target protein, consistent with its degradation upon 5-Ph-IAA treatment. This is indicated in the legends of Figures 1A, 3A, and S1A: “Asterisks denote putative truncated or fragmented forms of Cdu1.”

I previously requested that the authors state the availability of the materials described in the manuscript, such as plasmids and cell lines. However, I could not find this information in the revised version. As noted earlier, since this is a methodology paper, it is important to clearly indicate how these materials will be made available to the research community.

We have added a section called ‘Material Availability’ in the Methods section: “All plasmids, cell lines, bacterial strains, and chemical reagents generated or used in this study are available from the corresponding author upon request.”

Reviewer #2 (Remarks to the Author):

The authors have addressed the concerns about the *incA* targeting phenotype in the response to reviewers. In the response to reviewers comments the authors add more detail and analysis of the *incA* targeting experiments and their data and conclusions seem to be more inline with the published data on *incA* knockouts, knockdowns and natural mutants. This assuages the concerns about potential off target effects of their system.

We appreciate the reviewer’s conclusion on the specificity of the system.

However, the authors have not actually changed the wording of the manuscript. The authors mention that they have “Rather than referring to “fragmentation” or compromised “inclusion integrity,” we now focus specifically on homotypic inclusion

fusion phenotypes.” However, the abstract still states that “the integral membrane fusogen IncA is continuously required for inclusion integrity” and this is repeated throughout the discussion with statements such as “IncA-dependent steady state that demands continuous ICS-like IncA assemblies to counteract host-driven membrane destabilization, not solely on IncA’s role in initiating fusion” and IncA is continuously required for inclusion integrity.” These statements are not supported by the updated data.

We apologize; we have revised most of these passages but apparently overlooked a few in the first revision. We have now made the necessary changes.

Reviewer #3 (Remarks to the Author):

I thank the authors for conducting additional experiments and addressing my previous comments and concerns. The revised manuscript is thereby improved.

We thank the reviewer for the positive feedback.

A few points still require clarification regarding the role of IncA, which, as noted in the first round of review, represents an important and novel aspect of the manuscript.

- Statements indicating a role for IncA in “inclusion integrity” (lines 24, 451), “inclusion architecture” (line 445), “destabilizing the pathogen niche” (lines 456–457), and any other instances I may have missed should be removed. Additionally, the paragraph on lines 445–457 should be substantially revised to focus on the role of IncA in inclusion fusion/fission rather than inclusion integrity or stabilization.

We apologize for overlooking these passages in the initial revision and have now made the necessary changes.

- The live-imaging data presented in Figure 6 to monitor IncA-dependent inclusion fusion and fission are a welcome addition to the manuscript. However, it remains unclear what the authors intend the main take-home message of these observations to be. As noted above, both the Abstract and the Discussion emphasize inclusion stabilization, which is not directly supported by the data presented.

We agree that IncA plays a key role in inclusion fusion and fission. The passages in the initial revision referring to “inclusion stability” or “integrity” were imprecise and intended to describe the predominant phenotype in *C. trachomatis*-infected cells, namely the presence of a single inclusion. As noted above, we have revised these passages throughout the manuscript and now consistently refer to multi-inclusion phenotypes rather than (single) inclusion “integrity.”

We are therefore fully aligned with the view that IncA governs the transition between single inclusions (active IncA) and multiple inclusions (inactive IncA). This was our original intent, but it was not clearly articulated. All unclear passages have now been revised accordingly.

The key finding is that single inclusions convert into multiple inclusions in cells infected for 24 hours shortly after IncA depletion by auxin addition. Because inclusion fission can be directly observed under these conditions by live-cell imaging, we propose that IncA is required not only for the initial fusion event but also for maintaining the single-inclusion state thereafter.

- In the description of the still images in Figure 6C–D (lines 308–316), it would be more appropriate to describe the observations in terms of single versus multiple inclusions per cell, as reflected in the quantification shown in Figure 6D. At this stage, it is premature to conclude whether fusion or fission events are responsible for the observed number of inclusions per cell.

We agree with the reviewer and modified the text accordingly.

- Lines 308–309: the authors state that “control treatments establish the expected inclusion phenotypes... sustained IncA degradation caused inclusion fission.” As noted above, the phenotype should instead be described as multiple inclusions per cell. More importantly, a multiple-inclusion phenotype is not expected under the conditions described. Infection with an incA mutant strain at an MOI of 1 does not result in multiple inclusions per cell; the inclusion fusion defect of incA mutants becomes apparent primarily at MOIs greater than 1. Similarly, the statement that “continuous IncA expression maintained fused inclusions” is not appropriate if the infection was initiated at an MOI of 1, where a single inclusion per cell would already be expected.

We agree that the still images in Fig. 6C-D should not be interpreted as evidence of inclusion fission or fusion. We also confirm that the multiple-inclusion phenotype becomes more pronounced at higher MOI, which we have now validated experimentally. These data have been included as new figure S13 as requested by the reviewer.

However, we find no evidence in the literature that infection at an MOI of 1 results exclusively in single inclusions. For example, the Hackstadt group used an MOI of 1 in this study (<https://pmc.ncbi.nlm.nih.gov/articles/PMC4859576/>) but still observed multiple inclusions when using an IncA mutant. One possible explanation is that a subset of host cells becomes infected by more than one EB even at this MOI. In such cases, loss of IncA would be expected to impair homotypic inclusion fusion, leading to multiple inclusions per cell.

We have added the following clarifications to the manuscript:

Lines 288–290: “To remain consistent with previous studies of IncA function (47), following experiments were performed at MOI = 1.”

Lines 311–313: “...resulted in ~25% of infected cells containing multiple inclusions from 24 to 40 hpi, likely representing the subset of cells in which IncA-dependent homotypic fusion would normally occur (Treatment 1) (Fig. 6C, D).”

As noted above, we have also included additional MOI-dependent quantification in the revised manuscript (see Fig. S13 and lines 319–328).

I am not questioning the validity of the observed results. However, I would encourage the authors to:

- Avoid overinterpretation of the data (i.e., distinguishing between single/multiple inclusions versus fusion/fission events).
- Use more precise language, as a fusion event can only occur when multiple inclusions are present, and this should be reconciled with the experimental design using an MOI of 1.
- Clarify the central take-home message of the live-imaging experiments and what these results imply regarding the role of IncA.

These points have been partially addressed in the authors’ response to Reviewer 2, where the authors provide several of the clarifications I was looking for. The graphs referenced in that response should be incorporated into the manuscript, and the cited studies should be used to discuss the data more thoroughly.

We agree with the reviewer and have revised the text accordingly. In the updated manuscript, the endpoint analyses in Fig. 6C–D are now described strictly in terms of single versus multiple inclusions per infected cell, without inferring fusion or fission events from these static images. The terms *inclusion fission* and *inclusion fusion* are reserved exclusively for the live-cell imaging data presented in Fig. 6E and Supplementary Movies 1–12, where these dynamics can be directly observed. We have rewritten the corresponding Results section (lines 329–336) and added a clarifying statement in lines 334–336 to emphasize the specific mechanistic insight provided by the live-imaging experiments.

In addition, we have included a new paragraph describing experiments performed across different MOIs, with appropriate references to prior studies. The corresponding quantitative data are now presented in Supplementary Fig. 13 and incorporated into the Results section (lines 319–328). The cited studies referenced in our previous response to Reviewer 2 have now been discussed and incorporated into line 288-290, line 319-320, 327-328.